# OPEN

# CcrZ is a pneumococcal spatiotemporal cell cycle regulator that interacts with FtsZ and controls DNA replication by modulating the activity of DnaA

Clement Gallay [1,5], Stefano Sanselicio[1,5], Mary E. Anderson[2], Young Min Soh[1], Xue Liu[1], Gro A. Stamsås[3], Simone Pelliciari [4], Renske van Raaphorst[1], Julien Dénéréaz [1], Morten Kjos [3], Heath Murray [4], Stephan Gruber [1], Alan D. Grossman [2] and Jan-Willem Veening [1 ✉]

**Most bacteria replicate and segregate their DNA concomitantly while growing, before cell division takes place. How bacteria synchronize these different cell cycle events to ensure faithful chromosome inheritance by daughter cells is poorly understood. Here, we identify Cell Cycle Regulator protein interacting with FtsZ (CcrZ) as a conserved and essential protein in pneumococci and related Firmicutes such as *Bacillus subtilis* and *Staphylococcus aureus*. CcrZ couples cell division with DNA replication by controlling the activity of the master initiator of DNA replication, DnaA. The absence of CcrZ causes mis-timed and reduced initiation of DNA replication, which subsequently results in aberrant cell division. We show that CcrZ from *Streptococcus pneumoniae* interacts directly with the cytoskeleton protein FtsZ, which places CcrZ in the middle of the newborn cell where the DnaA-bound origin is positioned. This work uncovers a mechanism for control of the bacterial cell cycle in which CcrZ controls DnaA activity to ensure that the chromosome is replicated at the right time during the cell cycle.**

In many bacterial species, DNA replication and cell division occur concomitantly[1–3]. Different models exist to explain how *Escherichia coli* or *Bacillus subtilis* handle DNA replication together with cell division[4–7]. A current cell-size control model suggests that cells initiate DNA replication independently of their original size, and grow to a constant size independent of their size at birth (adder model)[8–12]. How cells sense changes in cell size and use this to trigger replication initiation is not known, but the models imply the existence of regulatory controls[3,13–15]. However, no such cell cycle regulator has been reported to date for bacteria.

Although *E. coli* and *B. subtilis* use different systems for regulating their cell cycles, the way they localize their division site is conserved, as both organisms use a variant of the Min system to prevent polymerization of the tubulin-like protein FtsZ away from the mid-cell[16,17]. Both species also have a nucleoid occlusion system (Noc) inhibiting Z-ring formation over the chromosome to prevent 'cutting' of the chromosome during cell division[18]. Together, the Min and Noc systems ensure that cell division and septation occur when both sister chromatids have been fully replicated and segregated. These systems are, however, not conserved in all bacteria because the Gram-positive opportunistic human pathogen *Streptococcus pneumoniae* lacks homologues of the Min and Noc systems[19]. In contrast to *E. coli* and *B. subtilis*, the pneumococcal Z-ring forms readily over the nucleoid[19,20]. Recently, a pneumococcal-specific protein called RocS was identified that might fulfil a similar function to the Noc system by connecting chromosome segregation and capsule production[21]. Another *S. pneumoniae*-specific protein called

MapZ was shown to guide Z-ring formation, analogous to the Min system in other bacteria[22–24]. Importantly, the position of the origin of replication (*oriC*) marks the approximate positions of future division sites[25]. In *S. pneumoniae*, cell division and DNA replication are thus intimately connected. However, it remains unknown how the cell senses when a new round of replication should be initiated.

We hypothesized that an unknown factor could be responsible for coordination of cell division and DNA replication in the pneumococcus. Using high-throughput gene silencing with clustered regularly interspaced short palindromic repeats interference (CRISPRi) of all essential genes of *S. pneumoniae*[26], we examined proteins leading to defects in DNA content on depletion. Here, we describe the identification of Cell Cycle Regulator protein interacting with FtsZ (CcrZ), a conserved protein that activates DnaA to trigger initiation of DNA replication. Pneumococcal CcrZ localizes at the division site in a FtsZ-dependent manner and its inactivation leads to division defects. Together, our findings show that CcrZ acts as a spatiotemporal link between cell division and DNA replication in *S. pneumoniae*.

## CcrZ is a conserved bacterial cell cycle protein

We previously generated a knockdown library using CRISPRi targeting 348 conditionally essential genes of the serotype 2 strain *S. pneumoniae* D39V that were identified by transposon-insertion sequencing[26]. Here, we investigated the function of *spv_0476*, encoding a protein of unknown function that is conserved in most Firmicutes (>30% identity) (Extended Data Fig. 1a). Silencing

[1]Department of Fundamental Microbiology, Faculty of Biology and Medicine, University of Lausanne, Lausanne, Switzerland. [2]Department of Biology, Massachusetts Institute of Technology, Cambridge, MA, USA. [3]Faculty of Chemistry, Biotechnology and Food Science, Norwegian University of Life Sciences, Ås, Norway. [4]Centre for Bacterial Cell Biology, Biosciences Institute, Newcastle University, Newcastle Upon Tyne, UK. [5]These authors contributed equally: Clement Gallay, Stefano Sanselicio. ✉e-mail: Jan-Willem.Veening@unil.ch

1175

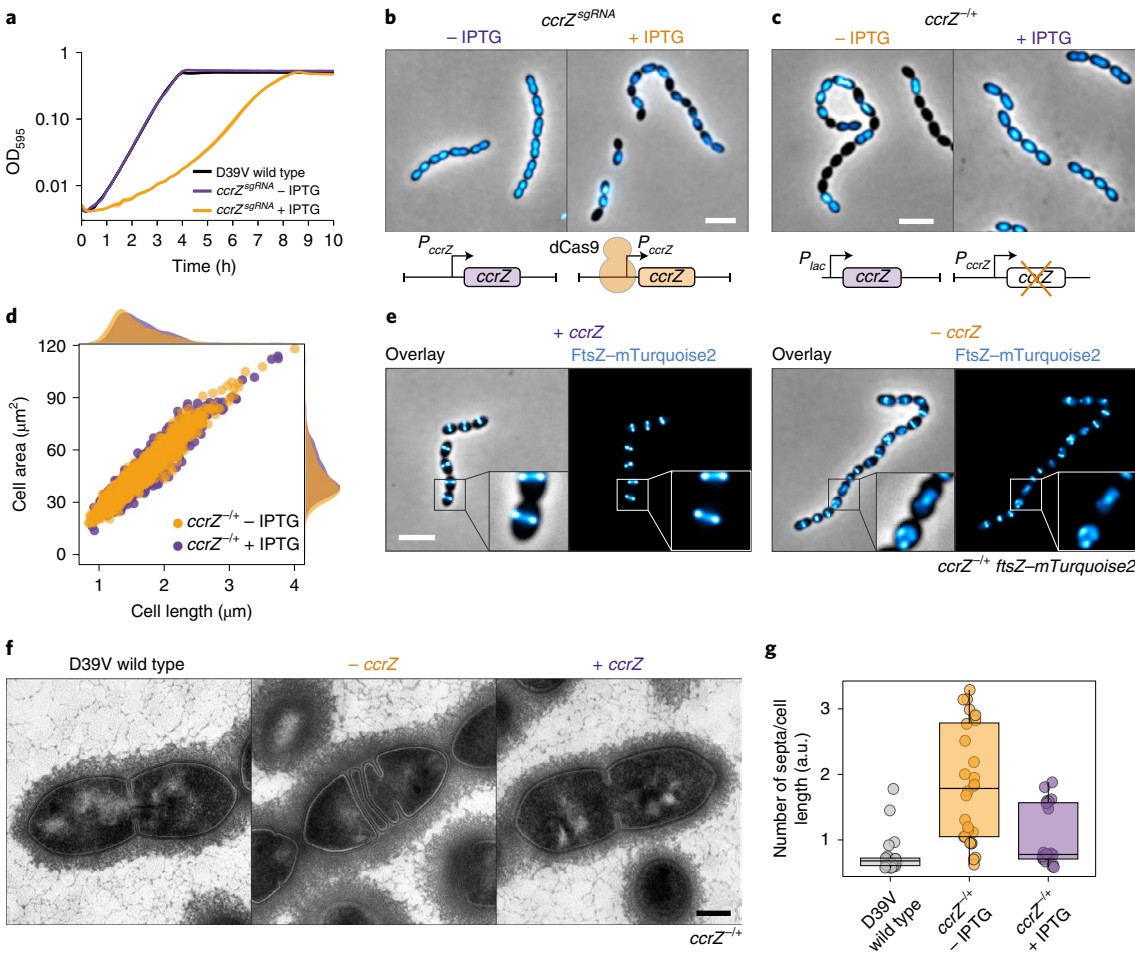

**Fig. 1 | Depletion of CcrZ leads to anucleate cells and division defects. a**, Growth curve of cells with *ccrZ* targeted by CRISPRi (*ccrZ*^sgRNA^ + IPTG) indicates a clear growth defect when *ccrZ* is silenced. **b**, *ccrZ* silencing leads to the appearance of anucleate cells, as visualized by DAPI staining. Scale bar, 3 μm. **c**, *ccrZ* depletion by ectopic expression via the IPTG-inducible $P_{lac}$ promoter also leads to cells lacking a nucleoid, as observed by DAPI staining. Scale bar, 3 μm. **d**, Distribution of cell area of *ccrZ*-depleted cells, *ccrZ* depletion leads to a slight decrease in cell length and cell area. $P = 1.5 \times 10^{-27}$ and $3 \times 10^{-22}$ when comparing length and area, respectively, values derived from two-sided Wilcoxon rank sum test. **e**, When a deletion of *ccrZ* is complemented (left), FtsZ–mTurquoise2 shows a clear mid-cell localization, whereas it appears as a blurry signal in several cells on *ccrZ* depletion (right). Scale bar, 3 μm. **f**, TEM indicates that cells depleted for *ccrZ* form multiple, often incomplete, septa. Scale bar, 250 nm. **g**, Distribution of number of septa per cell length as determined by TEM for $n = 22$ wild-type cells, $n = 28$ CcrZ-depleted cells and $n = 17$ CcrZ-complemented cells. Ratios are shown as box (25th to 75th percentile) and whisker (1.5× IQR) plots with individual data points overlaid as dot plots. $P = 1 \times 10^{-6}$ for wild-type versus *ccrZ*-depleted cells and $P = 0.0013$ for *ccrZ*-complemented versus *ccrZ*-depleted cells. $P$ values were derived from a two-sided pairwise Wilcoxon rank sum test with Bonferroni adjustment.

of *spv_0476* by CRISPRi led to a drastic reduction in growth rate as well as the appearance of anucleate cells, as visualized by 4,6-diamidino-2-phenylindole (DAPI) staining (Fig. 1a,b). We renamed SPV_0476 as CcrZ for reasons explained below. *ccrZ* is in an operon with *trmB*, which encodes a transfer RNA methyltransferase and this operon structure is conserved across Firmicutes (Extended Data Fig. 1a). To exclude the possibility that the observed phenotypes of *ccrZ* silencing were caused by polar effects on *trmB* expression, we constructed a deletion of *trmB*. This deletion did not lead to growth defects (Extended Data Fig. 1b). Transposon-insertion sequencing indicated that *ccrZ* is essential[26]; however, we were able to generate a Δ*ccrZ* deletion mutant, although cells grew slowly. We therefore constructed a depletion of CcrZ by ectopically expressing CcrZ under control of either an isopropyl-β-D-thiogalactoside (IPTG)- or a ZnCl₂-inducible promoter ($P_{lac}$ and $P_{Zn}$ respectively) and deleted *ccrZ* from its native location (*ccrZ*^−/+^ and $P_{Zn}$–*ccrZ*^−/+^ respectively). Depletion of CcrZ led to notable growth delay at 37 and 30 °C, confirming the

CRISPRi observations (Extended Data Fig. 1b). Immunoblotting using a specific antibody raised against purified CcrZ confirmed depletion (Extended Data Fig. 1c).

In line with the CRISPRi observations, DNA staining of cells depleted for CcrZ showed that 20% of cells lacked a nucleoid (Fig. 1c, $n = 442$ cells counted). To test whether the *ccrZ*-deletion phenotype was conserved in other Gram-positive bacteria, we silenced *ccrZ* (*SAOUHSC_01866*, here *ccrZ_Sa*) in *Staphylococcus aureus* SH1000 using CRISPRi and deleted the *B. subtilis* 168 *ccrZ* homologue (*ytmP*, here *ccrZ_Bs*). Upon *ccrZ_Sa* silencing in *S. aureus*, we observed a high proportion of anucleate cells, as well as a growth delay. By contrast, no anucleate cells were observed for *B. subtilis* (Extended Data Fig. 1d). However, cells deleted for *ccrZ_Bs* were slightly thinner and longer, although they had a growth rate similar to wild-type cells (Extended Data Fig. 1d). Interestingly, *S. pneumoniae* Δ*ccrZ* could not be complemented by expression of *ccrZ_Sa* or *ccrZ_Bs*. By contrast, depletion of *S. aureus* CcrZ was rescued by expression of CcrZ_Bs (Extended Data Fig. 1d).

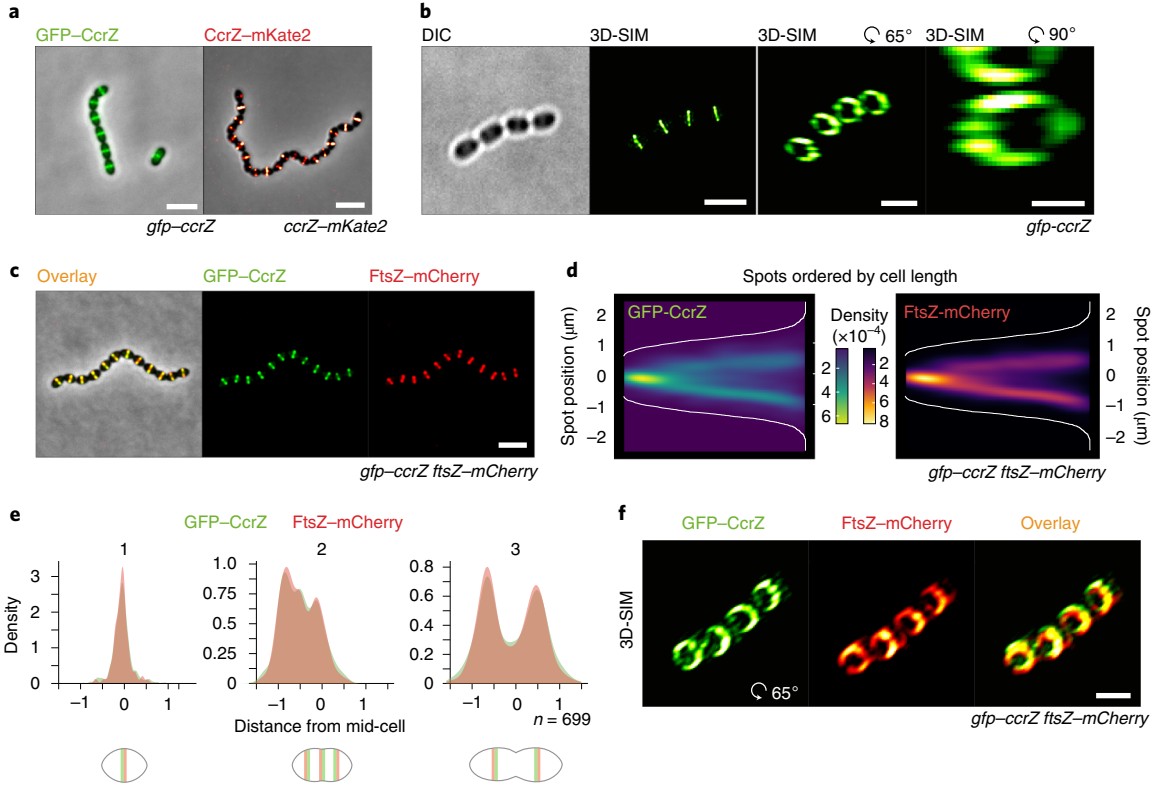

**Fig. 2 | CcrZ colocalizes with FtsZ at new division sites. a**, CcrZ localizes at mid-cell in live wild-type *S. pneumoniae* cells as observed by epifluorescence microscopy of GFP–CcrZ and CcrZ–mKate2. Scale bar, 3 μm. **b**, 3D-SIM of GFP–CcrZ and reconstructed volume projection (right) indicate that CcrZ forms a patchy ring at the mid-cell. Scale bar, 2 μm (left and middle right); 500 nm (right). **c**, GFP–CcrZ and FtsZ–mCherry colocalize in wild-type cells. Scale bar, 3 μm. **d**, Localization signal of GFP–CcrZ and FtsZ–mCherry in *n* = 699 cells of a double-labelled *gfp–ccrZ ftsZ–mCherry* strain, ordered by cell length and represented by a heatmap. **e**, GFP–CcrZ and FtsZ–mCherry colocalize during the entire cell cycle, as visualized when signal localization over cell length is grouped in three quantiles. **f**, 3D-SIM colocalization of GFP–CcrZ and FtsZ–mCherry shows a clear colocalizing ring with an identical patchy pattern. Note that for clarity, we did not correct for chromatic shift in the overlay. Scale bar, 2 μm.

CcrZ depletion in *S. pneumoniae* also led to slight morphological defects and modest changes in cell size when analysed by phase-contrast microscopy (Fig. 1d). Polysaccharide capsule production has previously been linked to the pneumococcal cell cycle[27], but capsule production was not impacted in a CcrZ mutant (Extended Data Fig. 1e). To visualize division sites in live cells, we constructed a translational fusion of mTurquoise2 to FtsZ (as the only copy of FtsZ, expressed from its native genetic location). As shown in Fig. 1e, Z-rings were clearly mislocalized upon CcrZ depletion for 3 h, with the presence of several aberrant Z-rings in 43% of the cells. To obtain further insight and verify that the increased number of septa are not due to the fluorescent protein fusion, we employed transmission electron microscopy (TEM) in untagged cells. Although not evident with phase-contrast microscopy, when *ccrZ* was depleted we observed frequent aberrant septum formation using TEM, and many cells harboured two (18%) to four (4%) septa, whereas only one septum is observed in 91% of wild-type cells (Fig. 1f,g).

### *S. pneumoniae* CcrZ is part of the divisome
Because CcrZ seems to be involved in both chromosome biology and cell division, we examined its subcellular localization. Strikingly, immunofluorescence on fixed cells using a CcrZ-specific antibody demonstrated a mid-cell localization (Extended Data Fig. 2a). To assess the localization of CcrZ in live cells, we created functional fusions of a green fluorescent protein (GFP) to the N terminus of CcrZ (*gfp–ccrZ*) or a red fluorescent protein to the

C terminus (*ccrZ–mKate2*) and inserted either construct at the native locus (Extended Data Fig. 1c). Epifluorescence microscopy of live bacteria showed that CcrZ localizes at mid-cell (Fig. 2a). This localization was conserved in both the TIGR4 and un-encapsulated R6 strains (Extended Data Fig. 2b). Interestingly, CcrZ$_{Sa}$ and CcrZ$_{Bs}$ did not localize as clear rings at mid-cell in *S. aureus* and *B. subtilis* (Extended Data Fig. 2b), indicating that activity and/or localization of CcrZ in these organisms is regulated differently. To obtain higher spatial resolution of *S. pneumoniae* GFP–CcrZ, 240 images (16 stacks) on live cells were acquired using three-dimensional structured illumination microscopy (3D-SIM) and reconstructed to generate a super-resolution image. As shown in Fig. 2b and Supplementary Video 1, CcrZ forms a patchy ring at mid-cell. Furthermore, time-lapse microscopy showed that CcrZ disassembles from the old septum to assemble at the newly formed division site (Supplementary Video 2).

To test whether the mid-cell localization of *S. pneumoniae* CcrZ coincides with FtsZ, we constructed a double-labelled strain (*gfp–ccrZ ftsZ–mCherry*). As shown in Fig. 2c, CcrZ colocalized with FtsZ (Fig. 2c–e and Supplementary Video 3). Note that the FtsZ–mCherry fusion did not affect growth or morphology[28]. 3D-SIM also showed a similar colocalizing pattern (Fig. 2f, Extended Data Fig. 2c and Supplementary Video 4).

Prediction of CcrZ's topology using TMHMM[29] did not indicate the presence of a transmembrane domain; the septal localization of CcrZ might then rely on another partner. To identify possible partners, we purified GFP–CcrZ expressed from *S. pneumoniae* and

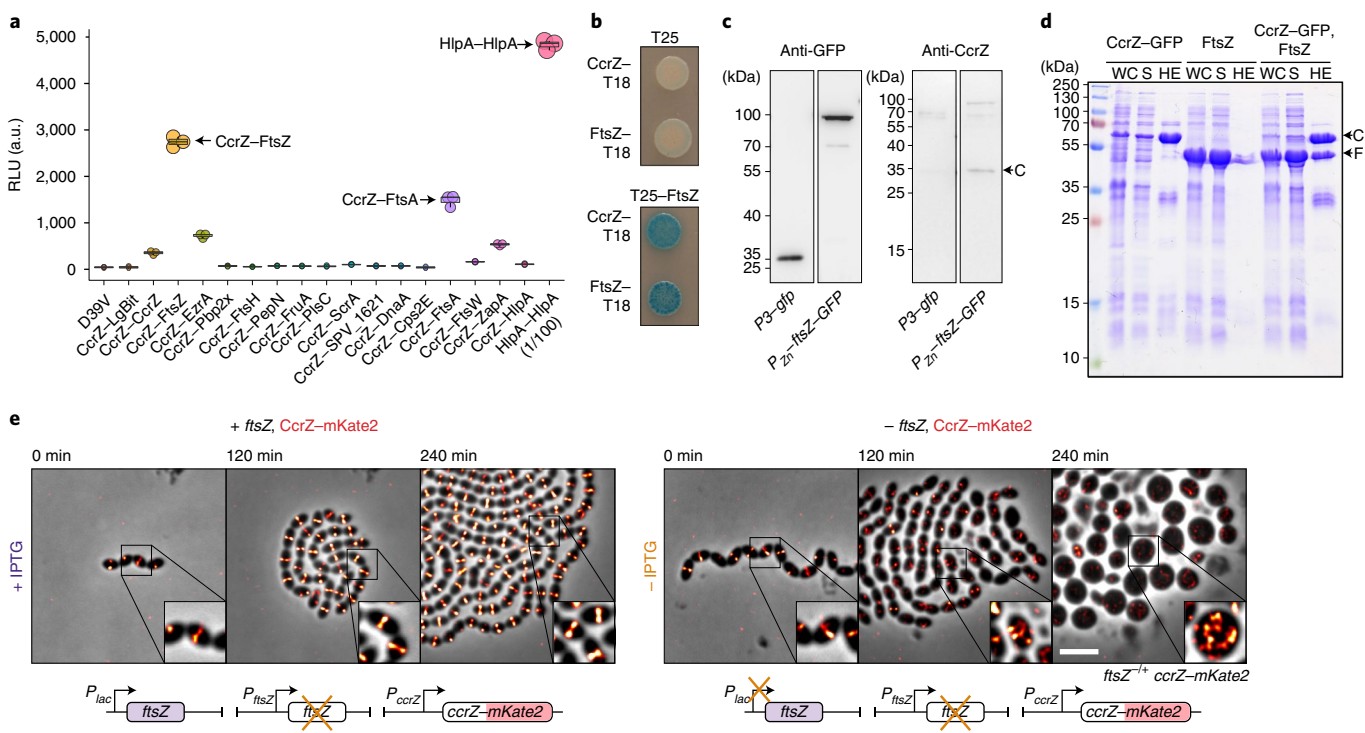

**Fig. 3 | CcrZ interacts directly with FtsZ. a**, Split-luciferase assay using several combinations with CcrZ–LgBit reveals that CcrZ and FtsZ are in very close proximity, as indicated by a high luminescence signal. FtsA, EzrA and ZapA, all three interacting directly with FtsZ, also gave a slight signal. *hlpA–LgBit hlpA–SmBit* (HlpA–HlpA), here diluted 100 times, is used as a positive control. Each dot represents the average of $n = 15$ measurements of a technical replicate, with the size of the dot representing the s.e.m. RLU, relative luminescence units. **b**, FtsZ–CcrZ interaction confirmation by bacterial two-hybrid assay. T25 is the empty vector pST25 and T25–FtsZ corresponds to vector pST25-FtsZ used in combination with pUT18-CcrZ (CcrZ–T18) and pUT18-FtsZ (FtsZ–T18). **c**, Affinity purification of FtsZ–GFP from *S. pneumoniae* cells (second lane) also pulls down untagged CcrZ (fourth lane). Purification of GFP alone (first lane) did not pull-down CcrZ (third lane). **d**, FtsZ from *S. pneumoniae* expressed in *E. coli* copurifies with CcrZ$_{Sp}$–GFP by affinity purification. WC, whole-cell extract; S, supernatant; HE, heat-eluted products; C, CcrZ–GFP; F, FtsZ. **e**, Epifluorescence time-lapse microscopy of CcrZ–mKate2 at 37 °C in the presence (left) or absence (right) of FtsZ. When FtsZ amounts are reduced, cells increase their size and CcrZ is delocalized from the mid-cell. Scale bar, 3 μm.

untagged cytosolic superfolder GFP as a control using anti-GFP nanobodies without cross-linking, and directly analysed the purified fraction using liquid chromatography tandem mass spectrometry. Interestingly, we found an enrichment (more than fivefold change) for several proteins from the divisome (for example, FtsZ, PBP2X and EzrA) (Supplementary Table 3). To determine which of the candidates might interact directly with CcrZ, we used the NanoBit complementation reporter assay[30,31], which uses a luciferase separated into a large bit (LgBit) and a small bit (SmBit). Fusion of two different interacting proteins to each fragment can restore the activity of the luciferase and, in the presence of a furimazine-based substrate, produce light[30]. Accordingly, we fused the C-terminal extremity of CcrZ to LgBit and putative partners to SmBit and integrated the different constructs at their respective loci under native control. We also fused SmBit to other proteins known to localize at the septum (Cps2E, FtsA, FtsW and ZapA). We used a strain expressing both HlpA–LgBit and HlpA–SmBit as a positive control for interaction[30]. After addition of substrate, we could detect a strong signal between FtsZ and CcrZ, as well as a weaker signal for FtsA, EzrA and ZapA, and no detectable signal for any of the other proteins (Fig. 3a). This result indicates that FtsZ and CcrZ in *S. pneumoniae* are in very close proximity in space. Interestingly, a weak signal was also observed for CcrZ–LgBit and CcrZ–SmBit (Fig. 3a).

To confirm the observed interaction with FtsZ, we used a bacterial two-hybrid assay in *E. coli*[32]. Again, we observed a robust interaction between CcrZ and FtsZ (Fig. 3b). Co-immunoprecipitation of FtsZ–GFP from *S. pneumoniae* cells confirmed the in vivo

interaction with CcrZ (Fig. 3c). Affinity purification of CcrZ$_{Sp}$–GFP when overexpressing FtsZ$_{Sp}$ in *E. coli* also confirmed this interaction because we were able to copurify FtsZ (Fig. 3d). To test whether the localization of CcrZ depends on FtsZ, we constructed a strain expressing CcrZ–mKate2 as well as a second copy of FtsZ under the control of an IPTG-inducible promoter and deleted the native *ftsZ* gene (*ftsZ*[−/+]). As expected, FtsZ depletion led to aberrant cell morphology and CcrZ–mKate2 was rapidly mislocalized (Fig. 3e and Supplementary Video 5). We conclude that CcrZ localizes to new division sites via a direct interaction with FtsZ.

## CcrZ controls DNA replication

To investigate the consequences of a lack of CcrZ on chromosome segregation in live cells, we introduced a translational fluorescent fusion of HlpA[33] and deleted *ccrZ*. Localization of HlpA–mKate2 in this slow-growing Δ*ccrZ* mutant showed similar results to DAPI-stained cells depleted for CcrZ and we observed that 19% of cells lacked a nucleoid signal (Extended Data Fig. 3a; 4,855 cells counted). Time-lapse imaging indicated that cells with defective DNA content had either no DNA at all or chromosomes 'guillotined' during septum closure suggesting reduced nucleoid occlusion control (Fig. 4a and Supplementary Video 6). We also colocalized FtsZ fused to a cyan fluorescent protein (FtsZ–CFP) with HlpA–mKate2 while depleting CcrZ for a short time (2 h). Interestingly, we observed many cells with a chromosome localized at only one half of the cell (Fig. 4b).

When attempting to make clean *ccrZ* deletions, in addition to small colonies typical of slow-growing mutants, there were also

spontaneous large, wild-type sized colonies. Growth analysis of three of these large colonies (*ccrZ*^supp1–3) showed that cells behaved like wild-types, and DAPI staining revealed normal DNA content (Fig. 4c,d). To verify whether these wild-type-like phenotypes were caused by suppressor mutations, we performed whole genome sequencing. All three strains still contained the *ccrZ* deletion and a single nucleotide polymorphism (Fig. 4e). Two missense mutations were found in *dnaA* (DnaA-Q247H and DnaA-S292G) and one nonsense mutation in *yabA* (YabA-E93*). Because DnaA promotes initiation of DNA replication and YabA hampers it by preventing interaction of DnaA with DnaN[34], we wondered whether the frequency of DNA replication initiation was changed in *a ccrZ* mutant.

To test this hypothesis, we quantified the copy-number ratio between chromosomal origin and terminus regions (*oriC/ter* ratios) using a real-time quantitative polymerase chain reaction (RT–qPCR). In a wild-type situation, during exponential growth, the *oriC/ter* ratio varies between 1.3 and 1.8, because most cells have started a round of replication (note that in contrast to *E. coli* and *B. subtilis*, multifork replication does not occur in *S. pneumoniae*)[35]. Remarkably, depletion of CcrZ resulted in a significantly decreased DNA replication initiation rate with an *oriC/ter* ratio of 1.1 versus 1.8 for complemented cells ($P < 0.05$) (Fig. 4f). Interestingly, the same observation was made for both *B. subtilis* and *Staphylococcus aureus*, where deletion or depletion of CcrZ caused a clear reduction in *oriC/ter* ratios (Fig. 4g,h). Because the identified *ccrZ*-bypass mutations were found in DNA replication initiation regulators, we tested whether they would restore the *oriC/ter* ratio in a fresh *ccrZ* deletion background in *S. pneumoniae*. Indeed, the *oriC/ter* ratios for Δ*ccrZ dnaA-S292G*, Δ*ccrZ dnaA-Q247H* and *yabA-E93** (*ccrZ*^supp3) were like wild-type (Fig. 4i,j).

The point mutation found in *yabA* causes premature translation termination at the C terminus of YabA. When *yabA* alone was replaced by an antibiotic resistance cassette, we observed an increase in replication initiation as well as a reduced growth rate; but when *ccrZ* was codeleted, wild-type-like growth and *oriC/ter* ratio was observed (Fig. 4j,k). DnaA suppressor mutations were located in the AAA+ ATPase domain of DnaA[36] (Extended Data Fig. 3b) and it was previously reported that specific mutations in this domain could increase the initiation rate in *B. subtilis*[37]. To determine whether those mutations alone were able to induce overinitiation, we inserted each *dnaA* mutation into a wild-type background. Marker frequency analysis detected an increase in the *oriC/ter* ratio for both *dnaA* alleles (Fig. 4l). We conclude that mutations that increase the rate of initiation of DNA replication can rescue the Δ*ccrZ* phenotype.

To gain additional insights into CcrZ function, we performed a genome-wide genetic interaction screen using CRISPRi-seq[38] (Fig. 4m). This technique relies on the expression of dCas9, controlled by an anhydrotetracycline (aTc)-inducible promoter, and constitutive

expression of a specific single guide RNA (sgRNA) that together form a roadblock for the RNA polymerase and thereby downregulate transcription of the targeted operon. We created a CRISPRi library by transforming *S. pneumoniae* $P_{tet}$–*dCas9*, $P_{lac}$–*ccrZ*, Δ*ccrZ* with 1,499 different sgRNAs targeting 2,111 of 2,146 genetic elements of *S. pneumoniae*. The resulting library was grown in the presence or absence of aTc/IPTG and the sgRNAs were Illumina sequenced. After analysing the fold change for every sgRNA between *ccrZ*-depletion and *ccrZ*-complementation, we found an enrichment of sgRNAs targeting the operon of YabA/HolB (*tmk-holB-yabA-spv_0828*), confirming that depletion of YabA can complement Δ*ccrZ* (Fig. 4n and Supplementary Table 4). Interestingly, we also found that inactivation of *ftsK* and *rocS* worsened the fitness of *a ccrZ* mutant. RocS is a regulator of chromosome segregation[21] and FtsK is important for chromosome segregation during cell division[39]. These interactions reinforce a role of CcrZ in chromosome integrity and replication and that CcrZ acts in a distinct pathway from these chromosome segregation factors. Finally, to test whether the mid-cell localization of CcrZ is important for well-timed replication of the chromosome, we abrogated CcrZ's mid-cell localization by depleting cells for FtsZ (Fig. 3e). After 2 h of FtsZ depletion, chromosomal DNA was isolated and *oriC/ter* ratios were determined. This showed that on mis-localization of CcrZ, cells under-replicate (Fig. 4o).

## CcrZ is a conserved regulator of DnaA

The results so far suggest that the division defects observed in the absence of CcrZ are caused by under-replication of the chromosome. To examine whether disruption of DNA replication in general could lead to defects in division, we took advantage of a thermosensitive *dnaA* mutant (*dnaA*^TS) in which DNA replication initiation is reduced when cells are grown at a non-permissive temperature (40 °C)[21]. As expected, when shifted to the non-permissive temperature, many cells were anucleate (Extended Data Fig. 4a). Strikingly, localization of FtsZ–mTurquoise2 in the *dnaA*^TS strain at 40 °C phenocopied the Δ*ccrZ* mutant, and FtsZ was frequently mislocalized (Fig. 5a). Examination by time-lapse microscopy following a temperature shift from 30 °C to 40 °C showed that FtsZ–mTurquoise2 mis-localization occurs after four to five generations (Supplementary Video 7). Furthermore, examination by TEM at 40 °C showed many cells with aberrant septa (Fig. 5b). These data are consistent with the idea that CcrZ exerts a control on DNA replication initiation.

To test whether CcrZ controls DNA replication by regulating DnaA activity, we made use of the fact that *a B. subtilis* Δ*ccrZ*$_{Bs}$ mutant also underinitiates (Fig. 4h) and a strain was constructed in which DNA replication was driven in a RepN-dependent manner (from a plasmid origin of replication *oriN*) rather than from DnaA-dependent initiation (from *oriC*). This showed no significant *ori/ter* ratio differences when *ccrZ* was deleted (Fig. 5c), suggesting

**Fig. 4 | CcrZ-depleted cells under replicate. a**, Time-lapse microscopy of HlpA–mKate2 at 30 °C in a Δ*ccrZ* mutant. Orange arrows indicate cells with no nucleoid; white arrows indicate cells with 'guillotined' DNA. **b**, FtsZ–CFP and HlpA–mKate2 colocalization on *ccrZ* depletion. **c**, Growth of three *ccrZ* suppressor mutants (*ccrZ*^supp1–3) compared with *ccrZ* depletion (orange). **d**, DAPI staining of the three suppressor mutants. **e**, Schematic representation of the suppressor mutations within DnaA domain III and within the YabA DnaA/DnaN binding motif (ANB). TM, tetramerization domain. **f,g,i,j,l,o**, *oriC/ter* ratios determined by RT–qPCR. Mean values are indicated under the boxes. Data from Monte Carlo simulations (Methods) are shown as box (25th to 75th percentile) and whisker (1.5× IQR) plots with outliers as dots. *P < 0.05, values calculated by bootstrap. **f**, *oriC/ter* ratios for D39V wild-type and *ccrZ* depleted cells; $n = 3$ independent samples. **g**, *oriC/ter* ratios for *S. aureus* on *ccrZ*$_{Sa}$ depletion; $n = 4$ independent samples. **h**, *oriC/ter* ratios on *ccrZ*$_{Bs}$ deletion in *B. subtilis* as determined by RT–qPCR. Mean values are indicated under the boxes. Data are shown as box (25th to 75th percentile) and whisker (1.5× IQR) plots. *P = 0.02 two-sided *t*-test; $n = 3$ independent samples. **i**, *oriC/ter* ratios of Δ*ccrZ* with *dnaA* suppressor mutations; $n = 3$ independent samples. **j**, *oriC/ter* ratios of Δ*yabA* Δ*ccrZ* and Δ*ccrZ yabA-E93**; $n = 3$ independent samples. **k**, Growth of Δ*yabA* Δ*ccrZ* and Δ*ccrZ dnaA-Q247H/S292G* compared with *ccrZ* depletion. **l**, *oriC/ter* ratios of *dnaA-Q247H* and *dnaA-S292G* in a wild-type background. **m**, Schematic overview of CRISPRi-seq. A library expressing 1,499 different sgRNAs targeting 2,111 genetic elements was grown while expressing or repressing dCas9. sgRNA counts after Illumina sequencing indicate which operons become more (beneficial) or less (detrimental) essential in *a ccrZ* depletion background. **n**, CRISPRi-seq of *ccrZ* depletion versus *ccrZ* expression shows a positive interaction between *ccrZ* and *yabA/holB* and a negative interaction between *ccrZ* and *ftsK/rocS*. **o**, *oriC/ter* ratios for FtsZ depletion (−IPTG) and complementation (+IPTG); $n = 12$ independent samples. Scale bars, 3 μm.

that CcrZ is an activator of DnaA-dependent initiation of replication in *B. subtilis*. We tested whether CcrZ interacts directly with DnaA and employed bacterial two-hybrid assays and the Split-luc system using pneumococcal CcrZ and DnaA (Fig. 3a and Extended Data Fig. 4b). However, none of these assays revealed a direct protein–protein interaction. Also, in line with our genetic data, we did not find a direct interaction of CcrZ with YabA, although YabA interacts with DnaA (Extended Data Fig. 4c). It is still possible that

CcrZ interacts directly with DnaA, but that we cannot detect it with these assays. Alternatively, another factor might be required for CcrZ's function or CcrZ indirectly affects the activity of DnaA.

## CcrZ's conserved residues are essential for its function

*S. pneumoniae* CcrZ is predicted to have a single aminoglycoside phosphotransferase enzyme family (APH) domain (Fig. 5d). Pairwise comparisons of profile-hidden Markov models (HMMs)

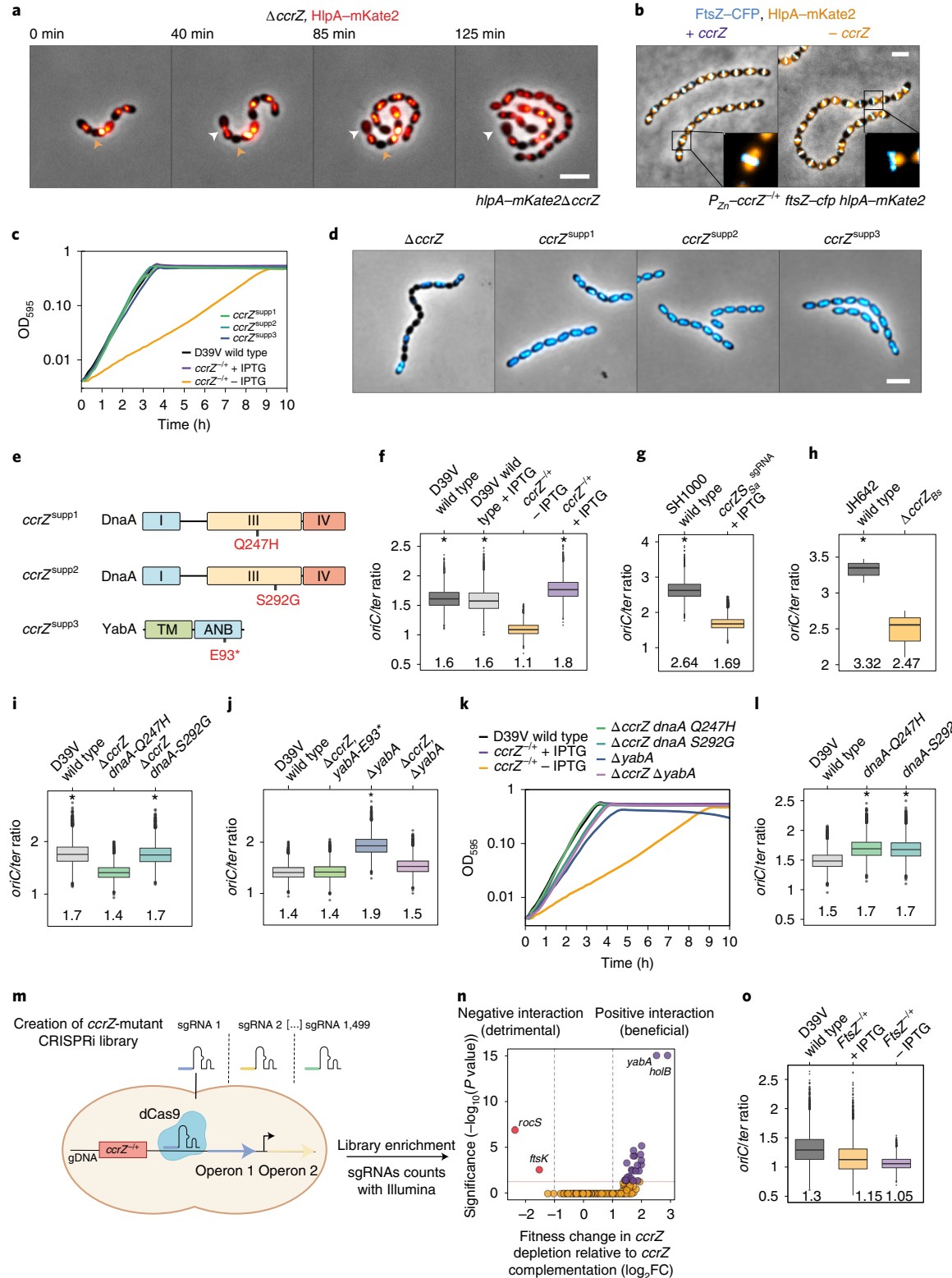

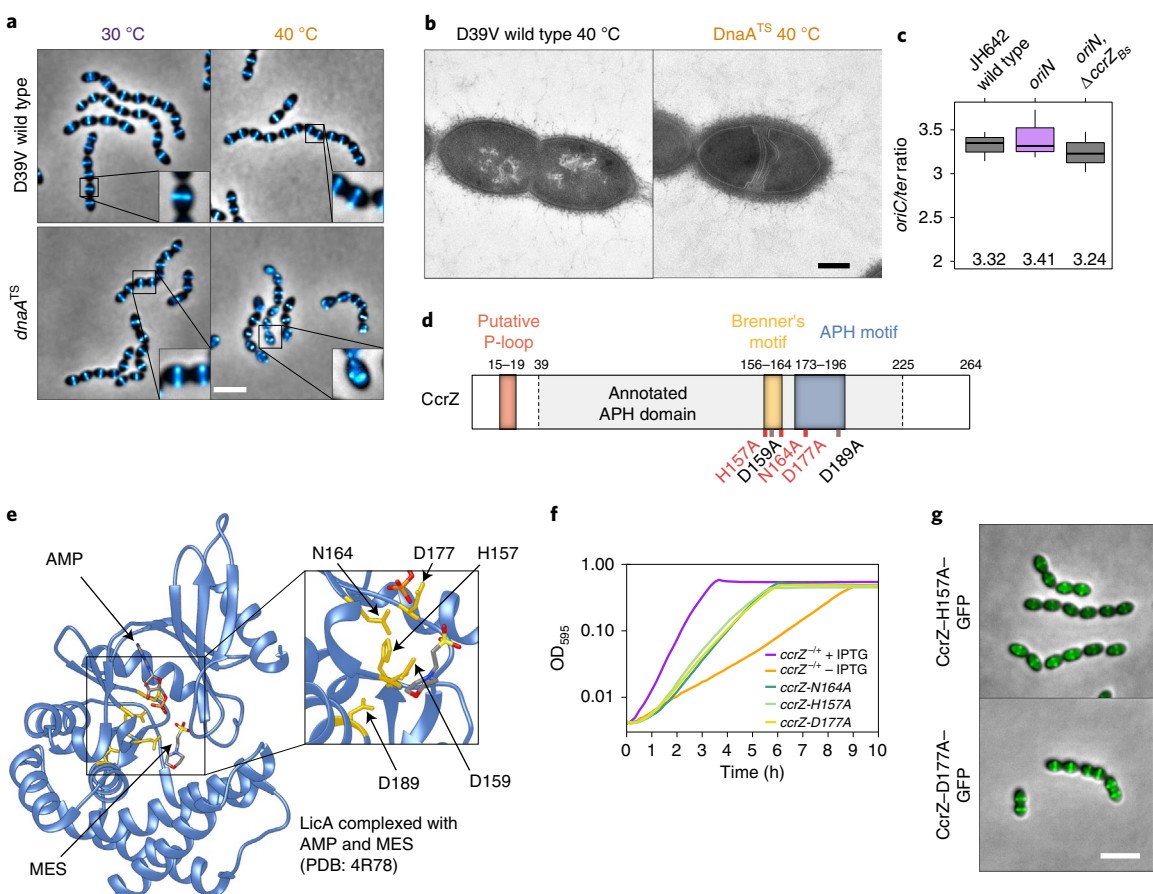

**Fig. 5 | CcrZ activates DnaA-dependent replication initiation. a**, Localization of FtsZ–mTurquoise2 in a thermosensitive DnaA strain (*dnaA*[TS]) at permissive (30 °C) and non-permissive (40 °C) temperatures shows that *dnaA* inactivation leads to a similar phenotype as *ccrZ* inactivation. Scale bar, 3 μm. **b**, TEM of DnaA[TS] at a non-permissive temperature (40 °C) indicates the presence of multiple septa, similar to a Δ*ccrZ* mutant. Scale bar, 250 nm. **c**, When replication is driven in a RepN-dependent manner in *B. subtilis* (*oriN*), no decrease in *ori/ter* ratio can be observed in the absence of *ccrZ*$_{Bs}$ (*oriN*, Δ*ccrZ*$_{Bs}$). Mean values are indicated under the boxes. Data are shown as box (25th to 75th percentile) and whisker (1.5× IQR) plots; *n* = 3 independent samples. **d**, Schematic representation of CcrZ motifs. CcrZ has one putative domain annotated APH (phosphotransferase enzyme family; PFAM01636). Sequence alignment with several kinases revealed the presence of a conserved P-loop, APH and Brenner's motifs, found in most phosphotransferases. Locations of mutations made for three essential (red) and two non-essential (black) conserved residues are shown underneath. **e**, LicA choline kinase structure complexed with AMP and MES buffer. The five residues indicated in yellow are conserved between CcrZ and LicA (and highly conserved within Firmicutes). **f**, Mutation of three of these five conserved residues in the putative ATP-binding pocket leads to growth defects. **g**, Localization of CcrZ–H157A–GFP and CcrZ–D177A–GFP is not impaired. Scale bar, 3 μm.

using HHpred[40] identified homologies with ethanolamine and choline kinases. Because CcrZ is highly conserved in Firmicutes, we aligned CcrZ with 1,000 sequences from UniRef50 and identified three residues conserved in more than 95% of the proteins (D159, N164 and D177) and two other residues (H157 and D189) conserved in more than 80% (Fig. 5d and Extended Data Fig. 4d). To determine the position of these residues, CcrZ was mapped onto the crystal structure of the best hit from the HMM alignment, the choline kinase LicA, in complex with adenosine monophosphate (AMP) (pdb 4R78). Interestingly, the five conserved residues are in spatial proximity to AMP (Fig. 5e). Comparison of CcrZ and LicA shows a conserved Brenner's motif [HXDhX3N] (residues CcrZ H157–N164) found in most phosphotransferases (Fig. 5d). In this motif, LicA–N187 (CcrZ–N164) was shown to interact with the α-phosphate moiety of AMP[41] and LicA–D176 (CcrZ–D159) was shown to be crucial for hydrogen bond formation with the hydroxyl moiety of choline. LicA–D194 (CcrZ–D177) interacts with the α-phosphate moiety of AMP. CcrZ, however, does not possess the conserved hydrophobic residues specific to choline and ethanolamine kinases necessary for choline binding, but instead

has several polar amino acids at these positions. Mutational analysis of the five conserved residues of CcrZ showed that at least H157, N164 and D177 are essential for CcrZ's function in *S. pneumoniae* (Fig. 5f), while mutating CcrZ–D159 or CcrZ–D189 did not lead to growth defects. All three essential mutants were properly produced (Extended Data Fig. 1c) and CcrZ–H157A and CcrZ–D177A could still localize at the septum (Fig. 5g). Given the high similarity with LicA, it is very likely that CcrZ can bind an as yet unknown nucleotide.

## A model for CcrZ-controlled DNA replication in *S. pneumoniae*

In *S. pneumoniae*, once DNA replication initiates at the mid-cell, the origins localize at both future division sites, while the replication machinery stays near the Z-ring until completion of replication and closure of the septum[25]. We therefore hypothesized that CcrZ is brought to the mid-cell by FtsZ to promote initiation of DNA replication. To map the hierarchy of events that take place during the pneumococcal cell cycle, we constructed a triple-labelled strain (strain *ccrZ–mKate2 dnaN–sfTQ*[OX] *parB*$_p$*–mYFP*) in which CcrZ is

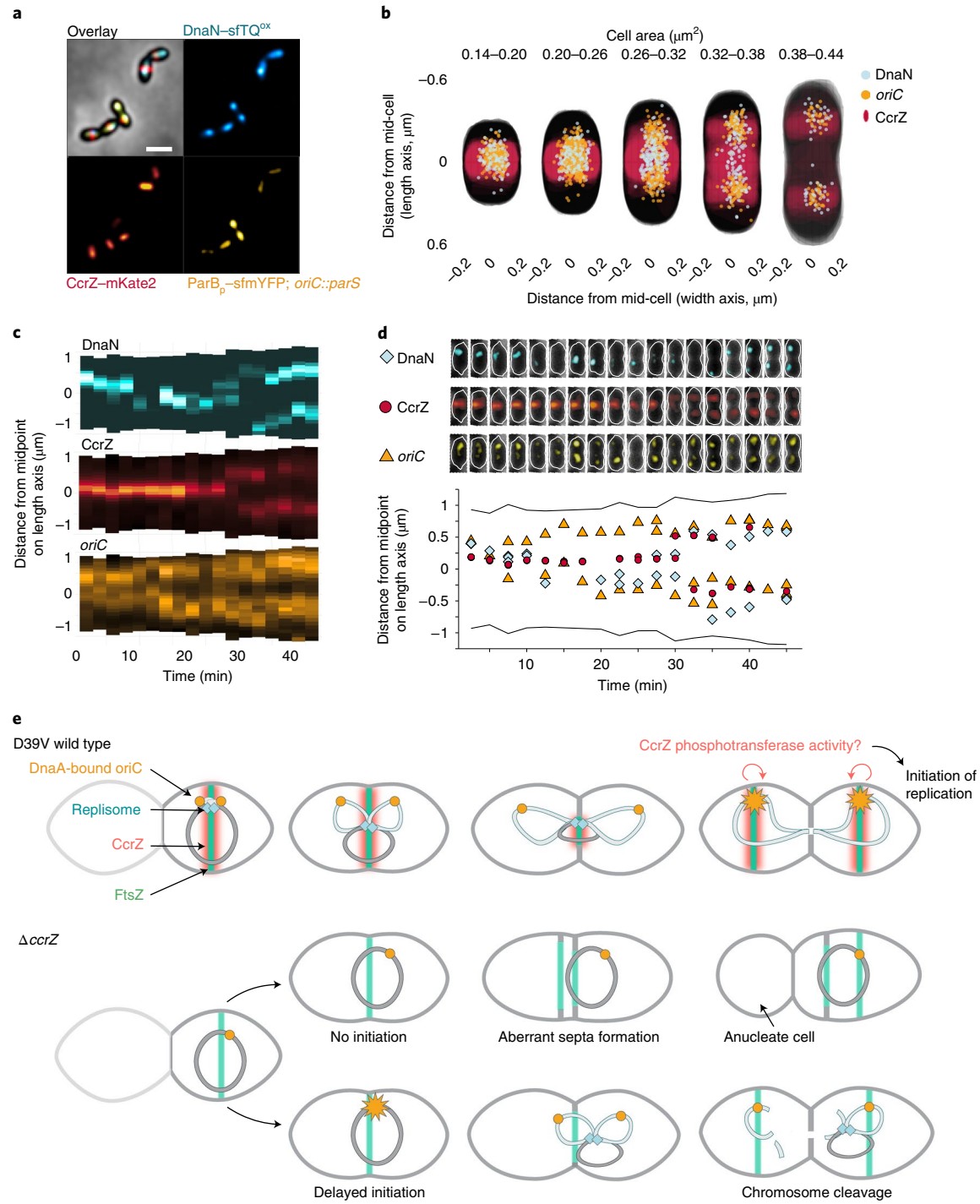

**Fig. 6 | Spatiotemporal localization of CcrZ via FtsZ ensures proper timing of DNA replication in *S. pneumoniae*. a**, Microscopy of the origin of replication (yellow), replication fork (cyan) and CcrZ (red) in live *S. pneumoniae* wild-type background cells. Scale bar, 1 μm. **b**, DnaN, *oriC* and CcrZ localizations grouped by cell area (μm²) in five equally sized groups. Analysed from snapshots of exponentially growing cells. **c**, Single-cell kymographs of DnaN, CcrZ and *oriC* localizations in a 2:30 minute interval time-lapse video. **d**, Tracked DnaN, *oriC* and CcrZ over time in a single cell. Top: overlay of fluorescence, cell outline and phase-contrast of the cell displayed in panel **c**. Bottom: fluorescence localization on the length axis of the same single cell over time. **e**, Model for spatiotemporal control of replication by CcrZ. In *S. pneumoniae*, CcrZ is brought to the middle of the cell where the DnaA-bound origin of replication is already positioned. CcrZ then stimulates DnaA to trigger DNA replication by an as yet unknown activity, possibly involving a phosphor-transfer event. Whereas the precise regulation and localization of CcrZ seems diverse between different organisms, CcrZ's activity to stimulate DNA replication is conserved, at least in *S. pneumoniae*, *S. aureus* and *B. subtilis*.

fused to a red fluorescent protein, DNA replication is visualized by DnaN fusion to a cyan fluorescent protein and the origin of replication is marked with a yellow fluorescent reporter. Imaging using time-lapse fluorescence microscopy revealed that DNA replication initiates once CcrZ is assembled at the mid-cell, rapidly followed by segregation of the newly replicated origins as cells elongate

(Fig. 6a–d and Supplementary Video 8). The replication machinery remains near the old division site together with CcrZ, moving to the new cell division sites only once DNA replication is complete. These data support a model in which FtsZ brings CcrZ to *oriC* to stimulate DnaA to fire a new round of replication ensuring that DNA replication commences only after the origins are well segregated and positioned at the new mid-cell position. Indeed, DnaA colocalizes with CcrZ in newborn cells (Extended Data Fig. 5). In the absence of CcrZ, initiation of DNA replication is mis-timed and occurs too late relative to cellular growth and Z-ring formation, frequently leading to futile division events, mis-segregated chromosomes and anucleate cells (Fig. 6e).

## Discussion

The principal contribution of this work is the identification and initial functional characterization of a mechanism for cell cycle regulation in *S. pneumoniae* via CcrZ. We show that CcrZ's septal localization occurs via direct interaction with FtsZ. Our data are consistent with a model in which, once positioned at mid-cell, CcrZ stimulates DnaA, probably by phosphorylation of an intermediate molecule, to initiate DNA replication (Fig. 6). Importantly, CcrZ's function of controlling DnaA seems conserved in *S. aureus* and *B. subtilis*, and probably in many other Gram-positive bacteria (Extended data Fig. 1a). In some ways, CcrZ resembles MipZ of *Caulobacter crescentus* in terms of one protein coordinating several aspects of the cell cycle. Whereas CcrZ stimulates replication after division has taken place, MipZ positions the Z-ring and delays division until chromosome segregation has initiated[42].

Besides the production of anucleate cells and cells with cleaved chromosomes, *ccrZ* mutants contain multiple aberrant division septa (Fig. 6e). Notably, this is phenocopied by a temperature-sensitive DnaA allele. This indicates that chromosome replication itself, and correct localization of the chromosome has an important role in nucleoid occlusion: when initiation is too late and the new daughter chromosomes are not fully segregated, division can take place over the DNA resulting in dissected chromosomes. We also observed multiple division septa in cells depleted for CcrZ that are probably caused by mis-timed chromosome segregation, whereby Z-rings are formed adjacent to the nucleoid. These phenotypes are reminiscent of observations made for *E. coli* and *B. subtilis* that showed that after arrest of DNA replication, many cells continued to elongate without dividing, but Z-rings continued to form to the side of nucleoids[43,44]. It is interesting to note that the *Staphylococcus aureus* Noc system also controls DNA replication, because Δ*noc* cells overinitiate DNA replication[45]. A lethal *S. aureus* Δ*noc* Δ*comEB* double mutant could be rescued by a suppressor mutation in *ccrZ*$_{Sa}$[45], further supporting that CcrZ$_{Sa}$ is also involved in the control of DNA replication.

This work uncovers a mechanism in which a single protein links cell division with DNA replication control. In this model, Z-ring formation is used as a timer for the initiation of DNA replication. When cell division terminates, leading to the formation of another Z-ring at the new division site, CcrZ is brought along and can activate a new round of DNA replication. This simple system ensures that DNA replication commences only once per cell cycle. It will be interesting to see how CcrZ controls the cell cycle in other bacteria, what the involved biochemical activities are and whether CcrZ will prove a new target for innovative antibiotics.

## Methods

**Bacterial strains and culture conditions.** All strains, plasmids and primers used are listed in Supplementary Table 1 and Supplementary Table 2.

All pneumococcal strains in this study are derivates of *S. pneumoniae* D39V[46], unless specified otherwise, and are listed in Supplementary Table 1. Strains were grown in liquid semidefined C+Y medium[47] at 37 °C from a starting optical density (OD$_{600}$) of 0.01 until the appropriate OD. Induction of the zinc-inducible promoter ($P_{Zn}$) was carried out by supplementing the medium with 0.1 mM ZnCl$_2$ and 0.01 mM MnCl$_2$ (Sigma-Aldrich) and the IPTG-inducible promoter ($P_{lac}$) was

activated with 1 mM IPTG (Sigma-Aldrich). For all related experiments, depletion strains were first grown without inducer until OD$_{600}$ = 0.3 and then diluted 100 times in fresh medium and grown until the desired OD. Transformation of *S. pneumoniae* was performed as described previously[47] with cells taken at the exponential growth phase (OD$_{600}$ = 0.1). When necessary, the medium was supplemented with the following antibiotics: chloramphenicol (0.45 μg ml⁻¹), erythromycin (0.2 μg ml⁻¹), kanamycin (250 μg ml⁻¹), spectinomycin (200 μg ml⁻¹) and tetracycline (0.5 μg ml⁻¹).

*Staphylococcus aureus* strains are listed in Supplementary Table 1. Cells were grown in brain heart infusion medium (Oxoid) with shaking at 37 °C. When appropriate, 5 μg ml⁻¹ of erythromycin and/or 10 μg ml⁻¹ of chloramphenicol was added to the growth medium. All *Staphylococcus aureus* plasmids were initially made in *E. coli* strain IM08B[48]. *E. coli* IM08B was grown in Luria–Bertani (LB) medium at 37 °C with shaking; 100 μg ml⁻¹ of ampicillin was added when appropriate. Plasmids were then transformed into *S. aureus* by electroporation, as described previously[49].

*B. subtilis* strains are listed in Supplementary Table 1. Cells were grown with shaking at 37 °C in LB medium or S7 defined minimal medium with MOPS buffer at a concentration of 50 mM rather than 100 mM supplemented with 1% glucose, 0.1% glutamate, trace metals, 40 μg ml⁻¹ phenylalanine and 40 μg ml⁻¹ tryptophan[50]. Standard concentrations of antibiotics were used when appropriate. *B. subtilis* strains were derived from 1A700 or JH642 (*pheA1 trpC2*)[51].

**Strain construction.** Construction of strains is described in the Supplementary Methods.

**Microtitre plate-based growth assay.** For *S. pneumoniae* growth assays, cells were first grown in C+Y medium pH 7.4 until the mid-exponential growth phase (OD$_{595}$ = 0.3) with no inducer at 37 °C, after which they were diluted 100 times in fresh C+Y medium supplemented with IPTG or ZnCl$_2$ when appropriate. Cellular growth was then monitored every 10 min at either 37 °C or 30 °C in a microtitre plate reader (TECAN Infinite F200 Pro). Each growth assay was performed in triplicate. The lowest OD$_{595}$ of each growth curve was normalized to 0.004 (detection limit of the reader and initial OD$_{595}$ of the inoculum) and the average of the triplicate values were plotted, with the s.e.m. represented by an area around the curve.

For assessment of *S. aureus* growth, CRISPRi knockdown strains were grown overnight in brain heart infusion medium. Cultures were then diluted 100-fold and grown until OD$_{600}$ = 0.4. The cultures were then re-diluted 200-fold in medium with or without inducer 500 μM IPTG. Growth analysis was performed on a Synergy H1 Hybrid (BioTek) microtiter plate reader at 37 °C with measurement of OD$_{600}$ every 10 min. The averages of the triplicate values were plotted, with the s.e.m. represented by an area around the curve.

**Phase-contrast and fluorescence microscopy.** *S. pneumoniae* cells were grown in C+Y medium pH 7.4 at 37 °C to an OD$_{595}$ = 0.1 without any inducer and diluted 100 times in fresh C+Y medium supplemented when appropriate with IPTG (for activation of dCas9, complementation of CcrZ and FtsZ, or expression of fluorescent fusions) or ZnCl$_2$ (for CcrZ complementation or expression of fluorescent fusions). At OD$_{595}$ = 0.1, 1 ml of culture was harvested by centrifugation for 1 min at 9,000g. For DAPI staining, 1 μg ml⁻¹ of DAPI (Sigma-Aldrich) was added to the cells and incubated for 5 min at room temperature before centrifugation. For imaging of bulk exponentially growing cultures, cells were washed twice with 1 ml of ice-cold PBS and resuspended in 50 μl of ice-cold PBS; for time-lapse microscopy, cells were washed and resuspended in 1 ml of fresh prewarmed C+Y medium. One microlitre of cells was then spotted onto PBS- or C+Y-polyacrylamide (10%) pads. For time-lapse microscopy, pads were incubated twice for 30 min in fresh C+Y medium at 37 °C before spotting. Pads were then placed inside a gene frame (Thermo Fisher Scientific) and sealed with a cover glass as described previously[52]. Microscopy acquisition was performed using either a Leica DMi8 microscope with a sCMOS DFC9000 (Leica) camera and a SOLA light engine (Lumencor) or a DV Elite microscope (GE Healthcare) with a sCMOS (PCO-edge) camera and a DV Trulight solid-state illumination module (GE Healthcare), and a ×100/1.40 oil-immersion objective. Phase-contrast images were acquired using transmission light (100 ms exposure). Still fluorescence images were usually acquired with 700 ms exposure, and time-lapses with 200–300 ms exposure. The Leica DMi8 filters set used were as followed: DAPI (Leica 11533333, Ex: 395/25 nm, BS: LP 425 nm, Em: BP 460/50 nm), CFP (Ex: 430/24 nm Chroma ET430/24x, BS: LP 455 Leica 11536022, Em: 470/24 nm Chroma ET470/24 m), GFP (Ex: 470/40 nm Chroma ET470/40x, BS: LP 498 Leica 11536022, Em: 520/40 nm Chroma ET520/40 m), yellow fluorescent protein (YFP) (Ex: 500/20 nm Chroma ET500/20x, BS: LP 520 Leica 11536022, Em: 535/30 nm Chroma ET535/30 m) and mCherry (Chroma 49017, Ex: 560/40 nm, BS: LP 590 nm, Em: LP 590 nm). DeltaVision microscope used a DV Quad-mCherry filter set: GFP (Ex: 475/28 nm, BS: 525/80 nm, Em: 523/36 nm) and mCherry (Ex: 575/25 nm, BS: 605/50, Em: 632/60 nm). Images were processed using either LasX v.3.4.2.18368 (Leica) or SoftWoRx v.7.0.0 (GE Healthcare). For *Staphylococcus aureus* microscopy, cells were induced as described above, grown until OD$_{600}$ = 0.2 and analysed on a Zeiss AxioObserver with an ORCA-Flash4.0 V2 Digital CMOS camera (Hamamatsu

 

Photonics) through a ×100 PC objective. An HPX 120 Illuminator (Zeiss) was used as a light source for fluorescence microscopy. Images were processed using ZEN Blue v.1.1.2.0 (Zeiss). Signals was deconvolved, when appropriate, using Huygens v.17.10.0p4 (SVI) software.

**Transmission electron microscopy.** Strains were grown in C+Y medium at either 37 °C, or at 30 °C for $dnaA^{TS}$, until $OD_{595} = 0.3$, with or without addition of $ZnCl_2$ (for $ccrZ$ complementation or depletion, respectively) and diluted 100 times in 10 ml of fresh C+Y medium. Cells were then grown at either 37 °C or 40 °C, for $dnaA$ depletion in the $dnaA^{TS}$ strain, until $OD_{595} = 0.15$. Five millilitres of each sample was then fixed with 2.5% glutaraldehyde solution (EMS) in phosphate buffer (PB 0.1 M pH 7.4) (Sigma-Aldrich) for 1 h at room temperature, followed by 16 h incubation at 4 °C. Cells were then postfixed using a fresh mixture of osmium tetroxide 1% (EMS) with 1.5% potassium ferrocyanide (Sigma-Aldrich) in PB for 2 h at room temperature. Samples were then washed three times with distilled water and spun down in low-melting agarose 2% (Sigma-Aldrich) and solidified in ice. Solid samples were cut in 1-mm³ cubes and dehydrated in acetone solution (Sigma-Aldrich) at graded concentrations (30% for 40 min; 50% for 40 min; 70% for 40 min and 100% for 3 × 1 h). This step was followed by infiltration in Epon (Sigma-Aldrich) at graded concentrations (Epon 1/3 acetone for 2 h; Epon 3/1 acetone for 2 h, Epon 1/1 for 4 h and Epon 1/1 for 12 h) and finally polymerized for 48 h at 60 °C. Ultra-thin sections of 50 nm were then cut on a Leica Ultracut (Leica Mikrosysteme) and placed on a copper slot grid 2 × 1 mm (EMS) coated with a polystyrene film (Sigma-Aldrich). Sections were subsequently poststained with 4% uranyl acetate (Sigma-Aldrich) for 10 min, rinsed several times with water, then with Reynolds lead citrate (Sigma-Aldrich) for 10 min and rinsed several times with distilled water. Micrographs were taken using a transmission electron microscope Philips CM100 (Thermo Fisher Scientific) equipped with a TVIPS TemCam-F416 digital camera (TVIPS) and using an acceleration voltage of 80 kV. Number of septa and cell length were measured manually on TEM images of cells in the correct focal plane: $n = 22$ wild-type cells, $n = 28$ $ccrZ$-depleted cells and $n = 17$ $ccrZ$-complemented cells.

**3D-structured illumination microscopy.** Samples for 3D-SIM were prepared as described previously by spotting 1 µl onto PBS–10% acrylamide pads. Acquisition was performed on a DeltaVision OMX SR microscope (GE Healthcare) equipped with a ×60/1.42 NA objective (Olympus) and 488 nm and 568 nm excitation lasers. Sixteen Z-sections of 0.125 µm each were acquired in Structure Illumination mode with 20 ms exposure and 20% laser power. The 240 images obtained were reconstructed with a Wiener constant of 0.01, and the volume reconstructed using SoftWoRx.

**Image analysis and cells segmentation.** All microscopy images were processed using FIJI v.1.52q (fiji.sc). Cell segmentation based on phase-contrast images was performed using either Oufti[53], MicrobeJ[54] or Morphometrics[55] and fluorescent signals where analysed using Oufti (for CcrZ and FtsZ), MicrobeJ[54] (for CcrZ) or iSBatch[56] (for DnaN and $oriC$). Fluorescence heat-maps were generated using BactMAP[57].

**Small-scale expression and GFP resin pull-down of FtsZ and CcrZ–GFP.** For affinity purification of $CcrZ_{Sp}$–GFP while expressing $FtsZ_{Sp}$, $ccrZ_{Sp}$ was amplified from D39V genomic DNA with primers 213/214 and the resulting fragment was assembled using Golden Gate allelic replacement strategy (BsaI) with plasmid pET-Gate2 ccdB (pSG436), pSG366, pSG367 and pSG2562, resulting in plasmid pSG2950. $ftsZ$ was amplified by PCR 215/216 on D39V genomic DNA and cloned into plasmid pJet1.2, resulting in plasmid pSG4227. The latter was then assembled with pSG1694 using Golden Gate assembly, leading to plasmid pSG4268. BL21 DE3 Gold competent cells were cotransformed with plasmids containing one each of *S. pneumoniae* FtsZ and CcrZ–GFP. Expression was ZYM-5052 autoinduction media[58]. Cells were sonicated in buffer containing 50 mM Tris pH 7.5, 150 mM potassium acetate, 5% glycerol and 5 mM β-mercaptoethanol (lysis buffer). Supernatant was then mixed with GFP resin which was produced by cross-linking nanobody[59] to NHS-Activated Sepharose 4 Fast Flow beads (GE Healthcare) according to the manufacturer's instructions. After 1 h of batch binding, resin was washed 10 column volume (CV) with lysis buffer. Beads were then resuspended in 50 µl of lysis buffer mixed with SDS–PAGE loading dye containing 5% w/v β-mercaptoethanol and heat treated at 95 °C for 15 min. Supernatant was collected and labelled heat elution samples. Whole-cell lysate, supernatant after sonication, and heat elution samples were loaded onto 15% SDS–PAGE gels and visualized by Coomassie staining.

**Large-scale purification of CcrZ–CPD for antibody production.** To express a fusion of *S. pneumoniae* CcrZ with a C-terminal cysteine protease domain, $ccrZ$ was amplified by PCR from D39V genomic DNA with primers 213/214 and assembled using Golden Gate allelic replacement strategy (BsaI) with plasmid pET-Gate2 ccdB (pSG436), pSG366, pSG367 and pSG2559. The resulting pSG2949 plasmid was then transformed into BL21 DE3 Gold cells using ZYM-5052 autoinduction media[58]. Cells were sonicated in buffer containing 300 mM NaCl, 50 mM Tris pH 7.5, 5 mM β-mercaptoethanol and protease inhibitor cocktail.

Supernatant was loaded onto a gravity flow column containing HisPur Cobalt Resin (Thermo Scientific). The column was washed 5 CV with buffer containing 100 mM NaCl, 20 mM Tris pH 7.5 and 5 mM β-mercaptoethanol. Because CcrZ had an affinity to the resin even without the cysteine protease domain, instead of on-column tag cleavage, elution was collected with buffer containing 150 mM imidazole, 100 mM NaCl, 20 mM Tris pH 7.5 and 5 mM β-mercaptoethanol, and tag cleavage was performed for 1 h at 4 °C by adding 1 mM inositol hexakisphosphate. The sample was further purified using a HitrapQ column and Superdex 200 16/600 pg column (GE Healthcare). The final storage buffer contained 100 mM NaCl, 20 mM Tris pH 7.5 and 1 mM dithiothreitol. For antibody production, sample was loaded onto a 15% SDS–PAGE gel. Edge wells were cut out and stained with Coomassie to determine the position of CcrZ on the gel. Gel portions containing CcrZ were sent for antibody production by Eurogentec.

**Western blot analysis.** Cells were grown in C+Y medium until $OD_{595} = 0.2$ and harvested by centrifugation at 8,000$g$ for 2 min at room temperature from 1 ml of culture. Cells were resuspended in 150 µl of Nuclei Lysis Solution (Promega) containing 0.05% SDS, 0.025% deoxycholate and 1% protease inhibitor cocktail (Sigma-Aldrich), and incubated at 37 °C for 20 min and at 80 °C for 5 min to lyse the cells. One volume of 4× SDS sample buffer (50 mM Tris–HCl pH 6.8, 2% SDS, 10% glycerol, 1% β-mercaptoethanol, 12.5 mM EDTA and 0.02% Bromophenol blue) was then added to three volumes of cell lysate sample and heated at 95 °C for 10 min. Protein samples were separated by SDS–PAGE (4–20%) and blotted onto polyvinylidene fluoride membranes (Merck Millipore). Membranes were blocked for 1 h with Tris-buffered saline (TBS) containing 0.1% Tween 20 (Sigma-Aldrich) and 5% dry milk and further incubated for 1 h with primary antibodies diluted in TBS, 0.1% Tween 20, 5% dry milk. Polyclonal CcrZ-antiserum concentration used was 1:5,000 and commercial polyclonal rabbit anti-GFP IgG (Invitrogen A-6455) was used at 1:5,000. Membranes were washed four times for 5 min in TBS, 0.1% Tween 20 and incubated for 1 h with secondary goat anti-rabbit IgG horseradish peroxidase-conjugated (Abcam AB205718) diluted 1:20,000 in TBS, 0.1% Tween 20 and 5% dry milk. Membranes were then washed four times for 5 min in TBS, 0.1% Tween 20 and revealed with Immobilon Western HRP Substrate (Merck Millipore).

**CcrZ–GFP purification with anti-GFP nanobodies.** $gfp$–$ccrZ$ and $P3$–$gfp$ (negative control) strains were grown in C+Y medium at 37 °C until $OD_{595} = 0.2$ and cells were harvested by centrifugation 15 min at 3,000$g$ at 4 °C. Cells were then incubated in sucrose buffer (0.1 M Tris–HCl pH 7.5, 2 mM $MgCl_2$, 1 M sucrose, 1% protease inhibitor cocktail (Sigma-Aldrich), 200 µg ml⁻¹ RNase A and 10 µg ml⁻¹ DNase (Sigma-Aldrich)) for 30 min at 30 °C, then incubated in hypotonic buffer (0.1 M Tris–HCl pH 7.5, 1 mM EDTA, 1% Triton, 1% protease inhibitor cocktail, 200 µg ml⁻¹ RNase A and 10 µg ml⁻¹ DNase) for 15 min at room temperature and cell debris were eliminated by centrifugation 30 min at 15,000$g$ and 4 °C. Cell lysate was then incubated with equilibrated GFP-Trap resin (Chromotek) at 4 °C for 2 h. After several washes with wash buffer (10 mM Tris–HCl pH 7.5, 150 mM NaCl, 0.5 mM EDTA, 1% protease inhibitor cocktail), beads were resuspended in 20 µl of 8 M urea, 50 mM triethylammonium bicarbonate, pH 8.0 and reduced with 5 mM dithiothreitol for 30 min at 37 °C. Cysteines were alkylated by adding 20 mM iodoacetamide and incubated for 30 min at room temperature in the dark. Samples were diluted 1:1 with triethylammonium bicarbonate buffer, digested by adding 0.1 µg of modified trypsin (Promega) and incubated overnight at 37 °C, followed by a second digestion for 2 h with the same amount of enzyme. The supernatant was collected, diluted twice with 0.1% formic acid and desalted on strong cation-exchange micro-tips (StageTips, Thermo Fisher Scientific) as described previously[60]. Peptides were eluted with 1.0 M ammonium acetate (100 µl). Dried samples were resuspended in 25 µl of 0.1% formic acid, 2% acetonitrile prior being subjected to nano liquid chromatography tandem mass spectrometry (LC–MS/MS).

**LC–MS/MS analysis.** Tryptic peptide mixtures (5 µl) were injected on a Dionex RSLC 3000 nanoHPLC system interfaced via a nanospray source to a high-resolution QExactive Plus mass spectrometer (Thermo Fisher Scientific). Peptides were separated on an Easy Spray $C_{18}$ PepMap nanocolumn (25 or 50 cm × 75 µm ID, 2 µm, 100 Å, Dionex) using a 35 min gradient from 4 to 76% acetonitrile in 0.1% formic acid for peptide separation (total time: 65 min). Full mass spectrometry (MS) survey scans were performed at 70,000 resolution. In data-dependent acquisition controlled by Xcalibur v.4.0.27.19 software (Thermo Fisher), the ten most intense multiply charged precursor ions detected in the full MS survey scan were selected for higher energy collision-induced dissociation (normalized collision energy = 27%) and analysis in the orbitrap at 17,500 resolution. The window for precursor isolation was of 1.6 $m/z$ units around the precursor and selected fragments were excluded for 60 s from further analysis.

MS data were analysed with Mascot v.2.5 (Matrix Science) set up to search the UniProt (www.uniprot.org) protein sequence database restricted to *S. pneumoniae* D39/NCTC 7466 taxonomy (339 SWISSPROT sequences + 1,586 TrEMBL sequences). Trypsin (cleavage at K,R) was used as the enzyme definition, allowing two missed cleavages. Mascot was searched with a parent ion tolerance of 10 ppm and a fragment ion mass tolerance of 0.02 Da (QExactive Plus). Iodoacetamide derivative of cysteine was specified in Mascot as a fixed modification. N-terminal

acetylation of protein, oxidation of methionine and phosphorylation of Ser, Thr, Tyr and His were specified as variable modifications. Scaffold software (v.4.4, Proteome Software) was used to validate MS/MS-based peptide and protein identifications, and to perform data set alignment. Peptide identifications were accepted if they could be established at >90.0% probability as specified by the Peptide Prophet algorithm[61] with Scaffold delta-mass correction. Protein identifications were accepted if they could be established at >95.0% probability and contained at least two identified peptides. Protein probabilities were assigned by the Protein Prophet algorithm[62]. Proteins that contained similar peptides and could not be differentiated based on MS/MS analysis alone were grouped to satisfy the principles of parsimony. Proteins sharing significant peptide evidence were grouped into clusters.

**Split-luciferase assay.** *S. pneumoniae* cells were grown in C+Y medium at 37 °C until OD$_{595}$ = 0.2 and washed once with fresh C+Y medium. 1% NanoGlo Live Cell substrate (Promega) was then added, and luminescence was measured 15 times at 37 °C every 30 s in a plate reader (TECAN Infinite F200 Pro). Measurements were performed in triplicate and the average values were plotted, with the s.e.m. represented by the dot size.

**Bacterial two-hybrid assay.** The bacterial two-hybrid assay was based on the method from Karimova et al.[32] with the following modifications. *dnaA*, *ccrZ* and *ftsZ* genes from *S. pneumoniae* D39V were cloned both into the low copy-number vector pUT18 and into the high copy-number vector pST25[63] using the enzymes BamHI and KpnI. *E. coli* strain HM1784 (BTH101 Δ*rnh::kan*) was transformed using each combination of plasmids. Chemically competent cells were incubated on ice for 60 min, heat shocking at 42 °C for 90 s and then inoculated at 37 °C in 3 ml of LB media supplemented with ampicillin (100 µg ml$^{-1}$) and spectinomycin (50 µg ml$^{-1}$) with mild agitation for 16 h. The OD$_{600}$ was adjusted to 0.5, cultures were diluted 1:1,000 and a 5-µl aliquot was spotted on a nutrient agar plate containing antibiotics (as above) containing 0.006% X-gal. Plates were incubated at 30 °C for 48 h and the images were captured using a digital camera.

**Co-immunoprecipitation of CcrZ and FtsZ–GFP with anti-GFP nanobodies.** *S. pneumoniae* cells were grown in C+Y medium at 37 °C until OD$_{595}$ = 0.2 and harvested by centrifugation 15 min at 3,000*g* at 4 °C. Cells were lysed using GFP–Trap_A Lysis buffer (Chromotek), 0.25% deoxycolate, 1% protease inhibitor cocktail incubated at room temperature for 10 min followed by incubation at 4 °C for 20 min. Cell lysate was incubated with equilibrated GFP-Trap resin (Chromotek) at 4 °C for 2 h. The resin was then washed three times in GFP-Trap_A Wash buffer (Chromotek) and GFP–proteins were eluted using SDS sample buffer at 95 °C for 10 min and analysed by immunoblotting.

**Genome resequencing of *ccrZ* suppressors by next-generation sequencing.** Strains *hlpA–mKate2* Δ*ccrZ*, *ccrZ*$^{supp1}$, *ccrZ*$^{supp2}$ and *ccrZ*$^{supp3}$ were grown in C+Y medium at 37 °C until OD$_{595}$ = 0.3 and cells were harvested by centrifugation for 1 min at 10,000*g*. Pellet was then resuspended into Nuclei Lysis Solution (Promega) containing 0.05% SDS, 0.025% deoxycholate and 200 µg ml$^{-1}$ RNase A at 37 °C for 20 min to lyse the cells and Protein Precipitation Solution (Promega) was added. DNA was then precipitated using isopropanol. The extracted genomes were then analysed by Illumina sequencing by GATC Biotech (Eurofins Genomics). Mutations were mapped onto D39V genome using breseq pipeline[64]. Genomes sequences are available at SRA (project PRJNA564501).

**oriC/ter ratios determination by RT–qPCR.** Determination of *S. pneumoniae* *oriC/ter* ratios was performed as follows. Cells were pregrown until OD$_{600}$ = 0.4 in C+Y medium at 37 °C, with or without inducer (ZnCl$_2$ or IPTG) for complementation and depletion conditions, respectively. Cells were then diluted 100 times in fresh C+Y medium supplemented when appropriate with inducer and harvested for genomic DNA isolation when they reached OD$_{600}$ = 0.1 (exponential phase). For normalization (*oriC/ter* ratio of 1), *dnaA* thermosensitive strain was grown for 2 h at non-permissive temperature (40 °C) in C+Y medium and harvested for chromosomal DNA isolation. As a negative (overinitiating) control, wild-type *S. pneumoniae* was incubated 2 h with 0.15 µg ml$^{-1}$ HPUra (DNA replication inhibitor) at 37 °C before harvesting. Primers pairs OT1/OT2 and OT3/OT4 were used to amplify the *oriC* and *ter* regions, respectively. Amplification by RT–qPCR was performed in triplicate using SYBR Select Master Mix (Applied Biosystems) on a StepOne Plus Real-Time PCR System (Applied Biosystems). For *S. aureus* *oriC/ter* ratio determination, overnight cultures were diluted 100-fold and grown until OD$_{600}$ = 0,4. These cultures were then re-diluted 200-fold in medium with 500 µM IPTG and grown until OD$_{600}$ = 0.2. As reference samples with an assumed *oriC/ter* ratio of 1, wild-type *S. aureus* SH1000 cells at OD$_{600}$ = 0.15 were supplemented with 50 µg ml$^{-1}$ rifampicin (inhibiting replication initiation) and incubated for 2 h for replication run-out. Cells were then harvested and lysed enzymatically by addition of 0.2 mg ml$^{-1}$ lysostaphin and 10 mg ml$^{-1}$ lysozyme, and genomic DNA was isolated using the Wizard Genomic DNA Purification Kit (Promega). Quantitative PCR (qPCR) reactions of 10 µl were set up with 5 µl of PowerUp™ SYBR Green Master Mix (Applied Biosystems), 500 nM of each primer OT5/OT and OT7/OT8 and 20 ng of DNA. In both cases, amplification

efficiencies of the primers and *oriC/ter* ratios were determined as described previously[35]. Data were plotted as a whisker plot in which whiskers represent the 25th and 75th percentiles of data from 10,000 Monte Carlo simulations, (*$P$ < 0.05, values caluculated by bootstrap). For determination of *B. subtilis* *oriC/ter* ratios, cultures were grown to the mid-exponential phase in LB medium, diluted back to OD$_{600}$ = 0.05 and grown to the mid-exponential phase (OD$_{600}$ = 0.2–0.4) at 37 °C. Cells were harvested in ice-cold methanol (1:1 ratio) and pelleted. Genomic DNA was isolated using Qiagen DNeasy kit with 40 µg ml$^{-1}$ lysozyme. The copy number of the origin (*oriC*) and terminus (*ter*) was quantified by qPCR to generate the *oriC/ter* ratio. qPCR was done using SSoAdvanced SYBR master mix and CFX96 Touch Real-Time PCR system (Bio-Rad). Primers used to quantify the origin region were OT9/OT10. Primers used to quantify the terminus region were OT11/OT12. Origin-to-terminus ratios were determined by dividing the number of copies (as indicated by the Cp values measured through qPCRs) of the origin by the number of copies quantified at the terminus. Ratios were normalized to the origin-to-terminus ratio of a temperature-sensitive mutant, *dnaB134* (KPL69), that was grown to have synchronized replication initiation, resulting in a 1:1 ratio of the *oriC/ter*. Data were plotted as whisker plots where whiskers represent the 25th and 75th percentiles of data and the *P* value was obtained by a two-tailed *t*-test.

**Genetic interactions determination by CRISPRi-seq.** The protocol for CRISPRi library construction, sequencing and analysis was performed as described previously[38]. Briefly, 1,499 plasmids containing a different sgRNA were transformed into strain P$_{tet}$-dCas9, P$_{lac}$-ccrZ, Δ*ccrZ* in the presence of 1 mM IPTG to ensure the expression of wild-type *ccrZ*, resulting in a pooled library containing the inducible CRISPRi system under control of an aTc-inducible promoter and combined with a depletion of *ccrZ* under control of an IPTG-inducible promoter. Colonies were harvested and stored at −80 °C. To ensure sufficient induction of the library, cells were grown for eight generations in triplicate. The pooled libraries were diluted 1:100 from stock in 10 ml of C+Y medium supplemented, or not, with 50 ng ml$^{-1}$ aTc and 1 mM IPTG and grown at 37 °C. At OD$_{600}$ = 0.4, cells were harvested and their genomic DNA isolated and prepared for MiniSeq (Illumina) sequencing with a custom sequencing protocol (www.veeninglab.com/crispri-seq). Significance of differential sgRNA depletion/enrichment between aTc treatments, across *ccrZ* complementation and depletion (absolute log$_2$FC > 1, FDR-adjusted *P* < 0.05) was tested with the DESeq2 package as described previously[38,65].

**Capsule immunofluorescence.** For fluorescence analysis of *S. pneumoniae* polysaccharide capsule, cells were grown in C+Y medium at 37 °C until OD$_{595}$ = 0.1 and 1:1,000 diluted serum anti-serotype 2 from rabbit (Neufeld antisera, Statens Serum Institut 16745) was added for 5 min at 4 °C. Cells were washed three times with fresh C+Y medium and 1 mg ml$^{-1}$ of superclonal recombinant secondary antibody goat anti-rabbit IgG coupled to Alexa Fluor 555 (Invitrogen A27039) was added for 5 min at 4 °C. Cells were then spotted onto a PBS–agarose slide. Acquisition of the fluorescent signal was performed on DV Elite microscope with mCherry filter set (Ex: 575/25 nm, BS: 605/50, Em: 632/60 nm).

**Conservation and gene neighbourhood.** The CcrZ protein sequence was aligned against all non-redundant protein sequences from NIH (ncbi.nlm.nih.gov) using PSI-BLAST for different Firmicutes families. Sequences with the highest identity were then aligned using Clustal Omega (ebi.ac.uk/Tools/msa/clustalo) and a phylogenetic tree was generated using Interactive Tree Of Life (itol.embl.de). Gene neighbourhood data were obtained from the STRING database (string-db.org). For residue conservation data, 1,000 sequences of CcrZ homologues were retrieved with PSI-BLAST (ebi.ac.uk) from the UniRef50 database. Conservation visualization was obtained using WebLogo 3 (weblogo.threeplusone.com). Sequences were then aligned using Clustal Omega and CcrZ sequence with conservation scores was mapped using UCSF Chimera (cgl.ucsf.edu/chimera) onto the crystal structure of *S. pneumoniae* LicA (PDB 4R78), the closest homologue protein using HMM–HMM comparison with HHpred20.

**Statistics and reproducibility.** Data analysis was performed using R (v.3.6.1). When comparing wild-type phenotypes with *ccrZ* depletion/complementation, a two-sided Wilcoxon rank sum test was used because we did not assume a normal distribution; some mutant cells can behave like wild-type because of the variable time of depletion or possible leakiness of P$_{lac}$ or P$_{Zn}$. When using box and whisker plots, the lower and upper quartiles, respectively, represent the 25th and 75th percentiles, the lower and upper whiskers represent, respectively, the 25th percentile + 1.5× interquartile range (IQR) and the 75th percentile + 1.5 IQR; the median is represented as a solid line. When plotting the *oriC/ter* ratios in Fig. 4, the outliers are also depicted by grey dots.

Data shown are represented as mean of at least three replicates ± s.e.m. if data came from one experiment with replicated measurement, and ± s.d. if data came from separate experiments. In general, each experiment shown in this paper was repeated independently at least two times with similar outcomes; *n* refers to the number of independent samples analysed per condition.

**Reporting Summary.** Further information on research design is available in the Nature Research Reporting Summary linked to this article.

## Data availability

The data that support the findings of this study are available from the corresponding author on request. Data gathered from string-db are available at https://string-db.org. Published crystal structure of LicA in complex with AMP is available at PDB 4R78. SPD_0476 (CcrZ) amino acid sequence can be found in the UniProt Knowledgebase (uniprot/A0A0H2ZQL5). Genomes sequences data are available at NCBI Sequence Read Archive (SRA) under the following accession number PRJNA564501 and CRISPRi-seq data are available under accession number PRJNA740244. Source data are provided with this paper.

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

## Acknowledgements

We appreciate the assistance from the Electron Microscopy Facility (EMF) and the Protein Analysis Facility (PAF) at the University of Lausanne (UNIL) and thank them for their support. We thank Wiep Klaas Smits (LUMC) for the Split-luc sequences and Tanneke den Blaauwen (UVA) for the mTQ^ox sequence before publication, Arnau Domenech (UNIL) for construction of strain *hlpA–LgBit hlpA–SmBit* and Zhian Salehian (NMBU) for help with cloning. Work in the Kjos laboratory is supported by a FRIMEDBIO grant (project number 250976) and a JPIAMR grant (project number 296906) from the Research Council of Norway. Work in the Murray laboratory was supported by a Wellcome Trust Senior Research Fellowship (204985/Z/16/Z) and a grant from the Biotechnology and Biological Sciences Research Council (BB/P018432/1). Work in the Grossman laboratory was supported, in part, by the National Institute of General Medical Sciences of the National Institutes of Health under award numbers R37 GM041934 and R35 GM122538. Work in the Veening laboratory is supported by the Swiss National Science Foundation (project grant 31003A_172861; NCCR 'AntiResist' 51NF40_180541; JPIAMR 40AR40_185533), a Novartis Foundation grant (#17B064) and ERC consolidator grant 771534-PneumoCaTChER.

## Author contributions

C.G. and J.-W.V. wrote the paper with input from all authors. C.G., S.S., M.E.A., Y.M.S., X.L., G.A.S., S.P., R.v.R., J.D. and M.K. performed the experiments. C.G., S.S., M.E.A., M.K., H.M., S.G., A.D.G. and J.-W.V designed, analysed and interpreted the data.

## Competing interests

The authors declare no competing interests.

## Additional information

**Extended data** are available for this paper at https://doi.org/10.1038/s41564-021-00949-1.

**Correspondence and requests for materials** should be addressed to J.-W.V.

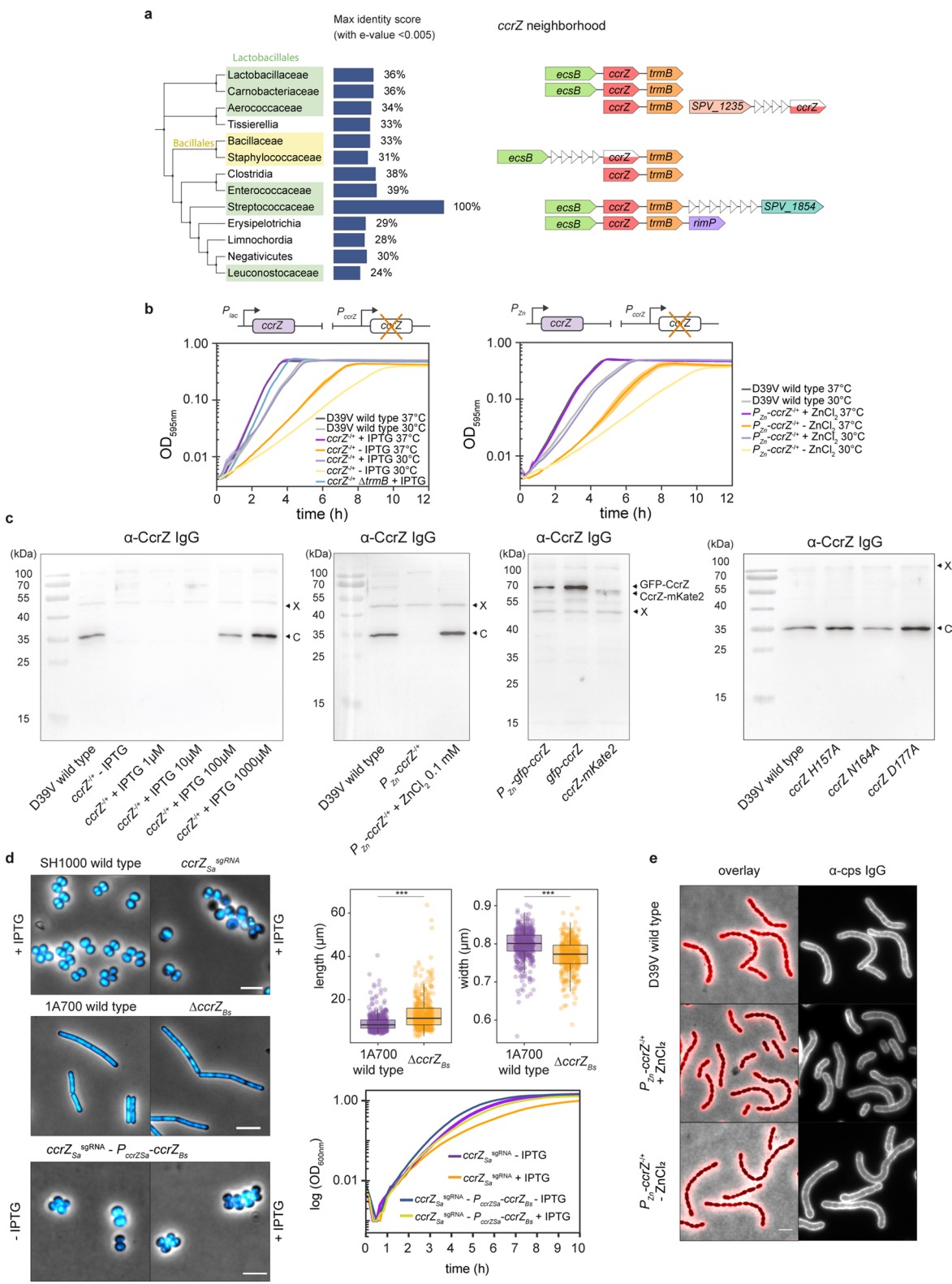

**Extended Data Fig. 1 | See next page for caption.**

**Extended Data Fig. 1 | *ccrZ* deletion phenotype is conserved in *S. aureus*. a**, Left: CcrZ conservation in firmicutes. Percentages indicate the highest percent identity, for each class, obtained using PSI-BLAST with NIH sequences. Right: genes co-occurrence in several genomes (data obtained from string-db.org; see Methods). Horizontal section indicate complexity in neighborhood score assignment; white triangles indicate missing annotation. **b**, Growth curves at 37 °C and 30 °C of *ccrZ* depletion mutants using $P_{lac}$ (left) or $P_{Zn}$ (right). **c**, Western blots for different pneumococcal strains. C: native CcrZ size; X: unknown protein recognized by α-ccrZ IgG. **d**, Microscopy of DAPI-stained *S. aureus* upon *ccrZ* silencing shows anucleate cells, while *B. subtilis* Δ*ccrZ* mutant did not present nucleoid defects. Top and bottom scale bars, 3 μm; middle scale bar, 5 μm. Δ*ccrZ* $_{Sa}$ cells were longer (or less well separated) and thinner (top right; wild type: n = 483 cells, Δ*ccrZ*$_{Bs}$ n = 399 cells; data are shown as box (25th to 75th percentile) and whiskers (1.5× IQR) plots with individual data points overlaid as dot plots) ***: *P* value = $6 \times 10^{-29}$ and *P* value = $2 \times 10^{-23}$ when comparing widths and lengths respectively, values derived from two-sided Wilcoxon rank sum test. Bottom-left: chromosome defects upon *ccrZ*$_{Sa}$ silencing can be rescued by expression of *ccrZ*$_{Bs}$ (*ccrZ*$_{Sa}$$^{sgRNA}$-$P_{ccrZSa}$-*ccrZ*$_{Bs}$ + IPTG). Associated growth curves (bottom-right) also confirmed the complementation of *ccrZ*$_{Sa}$ by *ccrZ*$_{Bs}$. **e**, Immunostaining of the polysaccharide capsule of *S. pneumoniae* wild type and upon *ccrZ* depletion. Scale bar, 3 μm.

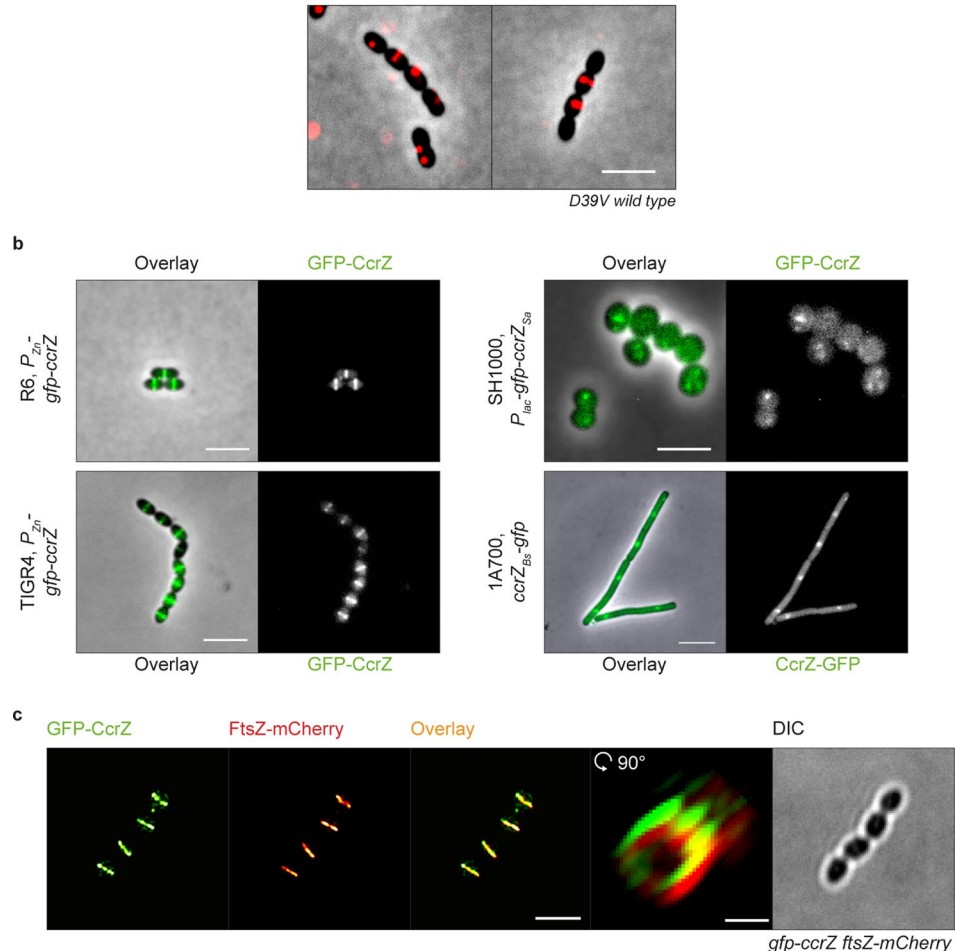

**Extended Data Fig. 2 | Septal localization of CcrZ in *S. pneumoniae*. a**, Immunostaining of CcrZ in wild type *S. pneumoniae* shows a septal localization. Scale bar, 3 μm. **b**, Localization of CcrZ in other pneumococcal strains (un-encapsulated R6 strain and capsular serotype 4 TIGR4) and in *S. aureus* SH1000, as well as in *B. subtilis* 1A700. Left and top-right scale bars, 3 μm; bottom-right scale bar, 5 μm. **c**, 3D-SIM of GFP-CcrZ (green) and FtsZ-mCherry (red) and reconstructed volume projection of both. Left scale bar, 2 μm; right scale bar, 500 nm.

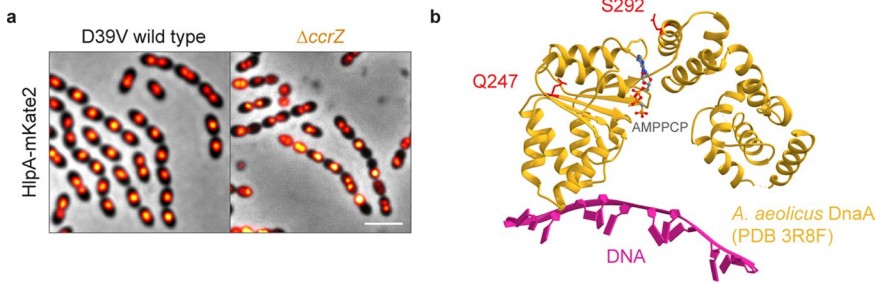

**Extended Data Fig. 3 | CcrZ controls DNA integrity. a**, Localization of the nucleoid associated protein HlpA in *a ccrZ* mutant shows anucleate cells. Scale bar, 3 μm. **b**, Mapping of DnaA Q247 and S292 residues onto the crystal structure of DnaA's AAA+ and duplex-DNA-binding domains from *Aquifex aeolicus*. Both residues are predicted to be in the AAA+ domain. DnaA Q247 and S292 correspond to *A. aeolicus* DnaA Q208 and E252 respectively.

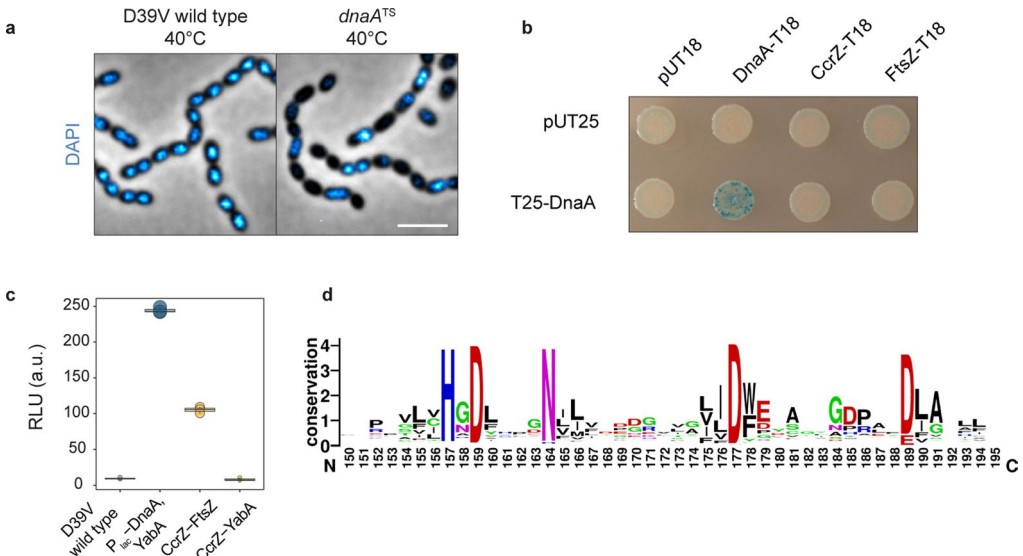

**Extended Data Fig. 4 | CcrZ activity is crucial for proper replication initiation as a dnaATS mutant phenocopied a *ccrZ* deletion. a**, Microscopy of DAPI-stained DnaA thermosensitive strain at non-permissive temperature (40 °C) indicates several anucleate cells, compared to a wild type grown in identical conditions. Scale bar, 3 μm. **b**, No interaction was detected between DnaA and CcrZ using bacterial-2-hybrid, while a positive DnaA-DnaA self-interaction is visible. **c**, Using split-luc assay, no interaction between CcrZ-YabA was detected, while a strong signal was obtained for DnaA-YabA. DnaA level was controlled by $P_{lac}$ to avoid toxicity. Each circle represents the average of n = 15 measurements of a technical replicate, with the size of the dot representing the SEM. **d**, Five (H157, D159, N164, D177 and D189) most conserved residues between 1,000 different CcrZ sequences from different bacterial species; sequences obtained from UniRef50 database.

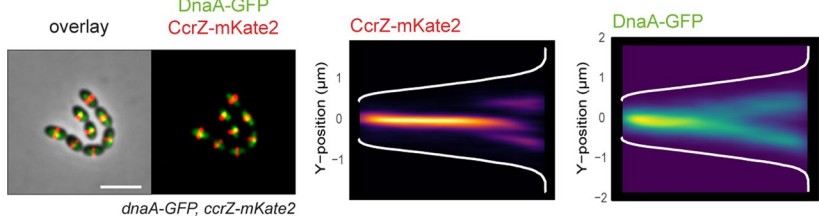

**Extended Data Fig. 5 | Transient co-localization of CcrZ and DnaA.** Co-localization of CcrZ-mKate2 with DnaA-GFP (left) and corresponding heatmap of signal distribution over cell length (right) show that DnaA and CcrZ co-localize at the beginning of the cell cycle. Scale bar, 3 μm.

# nature research

# Reporting Summary

Nature Research wishes to improve the reproducibility of the work that we publish. This form provides structure for consistency and transparency in reporting. For further information on Nature Research policies, see our Editorial Policies and the Editorial Policy Checklist.

## Statistics

For all statistical analyses, confirm that the following items are present in the figure legend, table legend, main text, or Methods section.

| n/a | Confirmed | |
|---|---|---|
| ☐ | ☒ | The exact sample size (*n*) for each experimental group/condition, given as a discrete number and unit of measurement |
| ☐ | ☒ | A statement on whether measurements were taken from distinct samples or whether the same sample was measured repeatedly |
| ☐ | ☒ | The statistical test(s) used AND whether they are one- or two-sided *Only common tests should be described solely by name; describe more complex techniques in the Methods section.* |
| ☒ | ☐ | A description of all covariates tested |
| ☐ | ☒ | A description of any assumptions or corrections, such as tests of normality and adjustment for multiple comparisons |
| ☐ | ☒ | A full description of the statistical parameters including central tendency (e.g. means) or other basic estimates (e.g. regression coefficient) AND variation (e.g. standard deviation) or associated estimates of uncertainty (e.g. confidence intervals) |
| ☐ | ☒ | For null hypothesis testing, the test statistic (e.g. *F*, *t*, *r*) with confidence intervals, effect sizes, degrees of freedom and *P* value noted *Give P values as exact values whenever suitable.* |
| ☐ | ☒ | For Bayesian analysis, information on the choice of priors and Markov chain Monte Carlo settings |
| ☒ | ☐ | For hierarchical and complex designs, identification of the appropriate level for tests and full reporting of outcomes |
| ☒ | ☐ | Estimates of effect sizes (e.g. Cohen's *d*, Pearson's *r*), indicating how they were calculated |

*Our web collection on statistics for biologists contains articles on many of the points above.*

## Software and code

Policy information about availability of computer code

| Data collection | OD measurement data where acquired using i-control 3.8.2.0 software (Tecan) or Gen5 3.04.17 (BioTek); Microscopy images were acquired using Las X 3.4.2.18368 (Leica), SoftWoRx 7.0.0 (GE Healthcare) or ZEN Blue 1.1.2.0 (Zeiss). |
|---|---|
| Data analysis | Microscopy images were deconvolved using Huygens 17.10.0p4 (SVI) and analyzed using Fiji 1.52q (ImageJ distribution; https://fiji.sc/). Microscopy signal was analyzed using Oufti (Paintdakhi, A. et al. 2016. Mol. Microbiol. 99, 767–777), MicrobeJ 5.13l (Ducret, A. et al. 2016. Nat Microbiol 1, 16077), Morphometrics (Ursell, T. et al. 2017. BMC Biol. 15, 17) and iSBatch (Caldas, V. E. A. et al. 2015. Mol. Biosyst. 11, 2699–2708). Mass spectrometry data were acquired using Xcalibur 4.0.27.19 (Thermo Fisher) and analyzed using Mascot 2.5 (Matrix Science) and Scaffold 4.4 (Proteome Software Inc). Data were analyzed and plotted using R 3.6.1. |

For manuscripts utilizing custom algorithms or software that are central to the research but not yet described in published literature, software must be made available to editors and reviewers. We strongly encourage code deposition in a community repository (e.g. GitHub). See the Nature Research guidelines for submitting code & software for further information.

## Data

Policy information about availability of data

All manuscripts must include a data availability statement. This statement should provide the following information, where applicable:

- Accession codes, unique identifiers, or web links for publicly available datasets
- A list of figures that have associated raw data
- A description of any restrictions on data availability

The data that support the findings of this study are available from the corresponding author upon request. Data gathered from string-db are available at https://string-db.org/cgi/network?taskId=bGMNPfTunceU&sessionId=bWgBIgywaw1D. Published crystal structure of LicA in complex with AMP is available at https://www.rcsb.org/structure/4R78. SPD_0476 (CcrZ) amino acid sequence can be found on the UniProt Knowledgebase https://www.uniprot.org/uniprot/A0A0H2ZQL5.

Genomes sequences data are available at NCBI Sequence Read Archive (SRA) under the following accession number PRJNA564501 and CRISPRi-seq data are available under accession number PRJNA740244.

# Field-specific reporting

Please select the one below that is the best fit for your research. If you are not sure, read the appropriate sections before making your selection.

☒ Life sciences ☐ Behavioural & social sciences ☐ Ecological, evolutionary & environmental sciences

For a reference copy of the document with all sections, see nature.com/documents/nr-reporting-summary-flat.pdf

# Life sciences study design

All studies must disclose on these points even when the disclosure is negative.

| | |
|---|---|
| Sample size | No sample-size calculations were performed. Sample sizes were chosen to allow appropriate statistical tests and were in line with other published studies in the field. |
| Data exclusions | No data was excluded. |
| Replication | All observed effects were highly significant and always successfully replicated (at least twice). |
| Randomization | Not applicable. Samples were not allocated to experimental groups |
| Blinding | Not applicable. Samples were not allocated to experimental groups |

# Reporting for specific materials, systems and methods

We require information from authors about some types of materials, experimental systems and methods used in many studies. Here, indicate whether each material, system or method listed is relevant to your study. If you are not sure if a list item applies to your research, read the appropriate section before selecting a response.

## Materials & experimental systems

| n/a | Involved in the study |
|---|---|
| ☐ | ☒ Antibodies |
| ☒ | ☐ Eukaryotic cell lines |
| ☒ | ☐ Palaeontology and archaeology |
| ☒ | ☐ Animals and other organisms |
| ☒ | ☐ Human research participants |
| ☒ | ☐ Clinical data |
| ☒ | ☐ Dual use research of concern |

## Methods

| n/a | Involved in the study |
|---|---|
| ☒ | ☐ ChIP-seq |
| ☒ | ☐ Flow cytometry |
| ☒ | ☐ MRI-based neuroimaging |

# Antibodies

| | |
|---|---|
| Antibodies used | Polyclonal rabbit anti-GFP IgG (Invitrogen #A-6455) ;<br>Goat anti-rabbit IgG HRP-conjugated (Abcam AB205718) ;<br>Serum anti-serotype 2 from rabbit (Neufeld antisera, Statens Serum Institut 16745) ;<br>Goat anti-rabbit IgG coupled to Alexa Fluor 555 (Invitrogen #A27039)<br>Anti-CcrZ IgG from rabbit serum |
| Validation | All four commercial antibodies (#A-6455,  #A27039, 16745, AB205718) were purchased from providers who have validated the antibodies for the use of Western blot and/or immunostaining .<br>Anti-CcrZ IgG were validated in the present study by Western blot with purified S. pneumoniae CcrZ. |

