## [Peer Review File · Nature Microbiology]

Peer Review Information

Journal: Nature Microbiology

Manuscript Title: CcrZ is a pneumococcal spatio-temporal cell cycle regulator that interacts with FtsZ and controls DNA replication by modulating the activity of DnaA

Corresponding author name(s): Jan-Willem Veening

Editorial Notes:

Redactions – transferred manuscripts (mention of previous referee reports from elsewhere)

Redactions – transferred manuscripts (mention of the other journal)

Redactions – unpublished data

This manuscript has been previously reviewed at another journal. This document only contains reviewer comments, rebuttal and decision letters for versions considered at Nature **XX**. Mentions of prior referee reports have been redacted

This manuscript has been previously reviewed at another journal. This document only contains reviewer comments, rebuttal and decision letters for versions considered at Nature **XX**. Mentions of the other journal have been redacted.

Reviewer Comments & Decisions:

Decision Letter, initial version:

Dear Jan-Willem,

Thank you for your patience while your revised manuscript "Spatio-temporal control of DNA replication by the pneumococcal cell cycle regulator CcrZ" was under peer-review at Nature Microbiology. It has now been seen by the 3 original referees {redacted}, whose expertise and comments you will find at the end of this email. As you will see, although they find your work of interest and appreciate the revisions made, they have still raised a few concerns that will need to be addressed before we can consider publication of the work in Nature Microbiology.

In particular, you will see that referees 1 and 3, although still remarking that the manuscript is missing a detailed molecular mechanism for how CcrZ regulates DNA replication, both indicate that they are satisfied with the revisions and support publication of the manuscript. Referee 2 makes similar comments about the lack of a detailed mechanism, but adds that it "is an interesting and novel story and CcrZ appears to be present in all Firmicutes so the findings should be of broad interest even without a mechanism." That said, this referee still raises several concerns that need to be addressed, particularly with regards to the interaction between FtsZ and CcrZ. As the referee states: "In this revised version they present data indicating this interaction depends upon the conserved tail of FtsZ (Fig. 3F). I have reservations about this finding although it would be of interest, if confirmed, since it would expand the number of proteins that interact with the tail of FtsZ. I just think it needs confirmation. Various publications have indicated that the interaction between the tail of FtsZ and its partners is extremely weak ($K_d \sim 30-50 \mu\text{M}$), which is too weak to mediate co-elution (Fig. 3d) (papers from the Lowe lab and Lutkenhaus lab). Also, in 3f (2-hybrid assay), there is presumably no Z ring (since only FtsZ deleted for the tail is expressed). If for some reason the Z ring is necessary for the functional interaction, it would not occur. If the interaction depends upon the tail of FtsZ this could be easily checked in vitro (they have the assay for this, just use an FtsZ with the tail deleted)." Therefore, we suggest that you further revise the manuscript to address this main point. The other comments from the referee seem clear and straightforward to address. Should further experimental data allow you to address these criticisms, we would be very happy to look at a revised manuscript.

We are committed to providing a fair and constructive peer-review process so please do not hesitate to contact us if there are specific requests from the reviewers that you believe are technically impossible or unlikely to yield a meaningful outcome.

We strongly support public availability of data. Please place the data used in your paper into a public data repository, if one exists, or alternatively, present the data as Source Data or Supplementary Information. If data can only be shared on request, please explain why in your Data Availability

Statement, and also in the correspondence with your editor. For some data types, deposition in a public repository is mandatory - more information on our data deposition policies and available repositories can be found at <https://www.nature.com/nature-research/editorial-policies/reporting-standards#availability-of-data>.

Please include a data availability statement as a separate section after Methods but before references, under the heading "Data Availability". This section should inform readers about the availability of the data used to support the conclusions of your study. This information includes accession codes to public repositories (data banks for protein, DNA or RNA sequences, microarray, proteomics data etc...), references to source data published alongside the paper, unique identifiers such as URLs to data repository entries, or data set DOIs, and any other statement about data availability. At a minimum, you should include the following statement: "The data that support the findings of this study are available from the corresponding author upon request", mentioning any restrictions on availability. If DOIs are provided, we also strongly encourage including these in the Reference list (authors, title, publisher (repository name), identifier, year). For more guidance on how to write this section please see: <http://www.nature.com/authors/policies/data/data-availability-statements-data-citations.pdf>

* If you have not done so already we suggest that you begin to revise your manuscript so that it conforms to our Article format instructions at <http://www.nature.com/nmicrobiol/info/final-submission>. Refer also to any guidelines provided in this letter.

{redacted}

Note: This url links to your confidential homepage and associated information about

manuscripts you may have submitted or be reviewing for us. If you wish to forward this e-mail to co- authors, please delete this link to your homepage first.

Nature Microbiology is committed to improving transparency in authorship. As part of our efforts in this direction, we are now requesting that all authors identified as 'corresponding author' on published papers create and link their Open Researcher and Contributor Identifier (ORCID) with their account on the Manuscript Tracking System (MTS), prior to acceptance. This applies to primary research papers only.

ORCID helps the scientific community achieve unambiguous attribution of all scholarly contributions. You can create and link your ORCID from the home page of the MTS by clicking on 'Modify my Springer Nature account'. For more information please visit www.springernature.com/orcid.

If you wish to submit a suitably revised manuscript we would hope to receive it within 3 months. If you cannot send it within this time, please let us know. We will be happy to consider your revision, even if a similar study has been accepted for publication at Nature Microbiology or published elsewhere (up to a maximum of 6 months).

In the meantime we hope that you find our referees' comments helpful and please don't hesitate to get in touch if you have any questions.

With best regards,

{REDACTED}

** Reviewer Comments:

Reviewer #1 (Remarks to the Author):

The authors have made a strong effort to address the my comments and those of the other reviewers, and have improved an already very good and important paper. Although the molecular mechanism by which CcrZ acts to regulate DNA replication is still unknown, I believe this version is worth publishing. I found no issues.

** Reviewer #2 (Remarks to the Author):

This is a revised manuscript describing a regulator of DNA replication (CcrZ) that is conserved in Firmicutes. The authors responded to most points raised by the reviewers, except the big one – some sort of molecular mechanism. It is very clear that deletion of CcrZ leads to under-replication of DNA which affects division and that can be suppressed by hyperactive mutations in DnaA or loss of the inhibitor of DnaA (YabA). The phenotypic changes due to loss of CcrZ are recapitulated by a DnaAts mutant. CcrZ colocalizes with FtsZ which functions to localize CcrZ and somehow regulate its activity. The data suggest that following initiation the new origins move to the ¼ and ¾ positions and await the arrival of FtsZ/CcrZ. How arrival of FtsZ/CcrZ licenses replication is still not clear despite efforts to find a direct link between CcrZ and DnaA or YabA. Based on sequence homology CcrA appears to be a

phosphotransferase. It is an interesting and novel story and CcrZ appears to be present in all Firmicutes so the findings should be of broad interest even without a mechanism.

One new piece of data is provided that I find confusing. Previously, the authors show strong evidence that FtsZ and CcrZ interact. In this revised version they present data indicating this interaction depends upon the conserved tail of FtsZ (Fig. 3F). I have reservations about this finding although it would be of interest, if confirmed, since it would expand the number of proteins that interact with the tail of FtsZ. I just think it needs confirmation. Various publications have indicated that the interaction between the tail of FtsZ and its partners is extremely weak ($K_d \sim 30-50 \mu\text{M}$), which is too weak to mediate co-elution (Fig. 3d) (papers from the Lowe lab and Lutkenhaus lab). Also, in 3f (2-hybrid assay), there is presumably no Z ring (since only FtsZ deleted for the tail is expressed). If for some reason the Z ring is necessary for the functional interaction, it would not occur. If the interaction depends upon the tail of FtsZ this could be easily checked in vitro (they have the assay for this, just use an FtsZ with the tail deleted).

Minor comments:

1. A lot of fluorescent fusions are used that are inserted in the chromosome in place of the WT gene. It seems that these all complement, although it is not specifically stated; e.g. FtsZ-mCherry in Fig. 2. It should be mentioned that cells are normal and the fusions complement (they look like it).
2. The suppression of the various phenotypes due to loss of CcrZ by DnaA-S292G and YabA deletion appears to be very good. Are there any noticeable defects in the absence of the CcrZ regulation?
3. Line 138. 1:1 ratio. I did not see this data in Fig. 3 or the supplement. I would like to see the sizing column data on the 1:1 ratio of FtsZ to CcrZ; is the size as expected for 1 FtsZ and 1 CcrZ. If the interaction is mediated by the tail of FtsZ this could readily be checked in vitro.
4. Line 214. Curious if the starting slow growing strain was sequenced? It may have suppressors that aid its growth without producing large colonies.
5. Line 230. I find it surprising that deletion of CcrZ from *B. subtilis* affected the ori/ter ratio but had no effect on morphology, Fig. 1d in extended data.
6. Line 329. Since you have the purified protein have you checked by mass spec to see if there is a bound nucleotide?

** Reviewer #3 (Remarks to the Author):

The authors have conducted additional experiments in response to my comments and provided additional information regarding the potential mechanism of how CcrZ regulates DanA protein. Although the mechanism has not been completely clarified, the authors have made effort in their current capacity to gain further insight. I find that the revision is satisfactory and that the manuscript has been

significantly improved that its publication in Nature Microbiology is

warranted. Hisao Masai

Author Rebuttal to Initial comments

We greatly appreciate the comments from all referees, and were very pleased to learn referees #1 and #3 are supportive of the publication and referee #2 only had a minor remaining comment. Nevertheless, we took this last point very serious and performed a set of new experiments (see below). The comment was that it would be quite surprising if indeed FtsZ would interact with CcrZ via its C-terminal tail (FtsZ-CTD).

This concern is pertinent since, as stated by reviewer #2, only a small set of proteins interact via this domain. We have performed additional experiments and, even though we consistently see a reduction of interaction between CcrZ and FtsZ Δ CTD *in vivo* using a SplitLuc assay, we found that the truncated FtsZ still interacts with CcrZ in a bacterial two-hybrid system, and we could still pull-down FtsZ Δ CTD when purifying CcrZ-GFP *in vitro*. Because of these new data, we now conclude that CcrZ does **not** interact, or partially, with FtsZ C-terminal tail and we hypothesized that the original reduction of interaction observed *in vivo* might in fact come from rapid active degradation of FtsZ Δ CTD in the cell. We have therefore removed the original claim from the manuscript.

Another suggestion from reviewer #2 was to look into more detail into the potential morphological changes of *B. subtilis* in absence of *ccrZ*. We have therefore performed further characterizations of a Δ *ccrZ*_{Bs} and in fact observed an increase in cell length and decrease in cell width compared to the wild type. In addition, we find that the Δ *ccrZ*_{Bs} is slightly more susceptible to ciprofloxacin. Finally, we found that CcrZ_{Bs} can bypass the lethality effect of a *yabA/dnaA1* hyper-initiating mutant (Anderson and Grossman, unpublished). Together, this data strongly suggests that Δ *ccrZ*_{Bs} has a similar function as *S. pneumoniae* and *S. aureus* CcrZ, although it is not essential on its own. We have added some of these results to the manuscript and improved the text with the other minor comments from reviewer #2.

Specific point-by-point responses to reviewer #2's comments and other changes made to the manuscript follow.

Reviewer #1 (Remarks to the Author):

The authors have made a strong effort to address my comments and those of the other reviewers and have improved an already very good and important paper. Although the molecular mechanism by which CcrZ acts to regulate DNA replication is still unknown, I believe this version is worth publishing. I found no issues.

Reviewer #2 (Remarks to the Author):

This is a revised manuscript describing a regulator of DNA replication (CcrZ) that is conserved in Firmicutes. The authors responded to most points raised by the reviewers, except the big one – some sort of molecular mechanism. It is very clear that deletion of CcrZ leads to under-replication of DNA which affects division and that can be suppressed by hyperactive mutations in DnaA or loss of the inhibitor of DnaA (YabA). The phenotypic changes due to loss of CcrZ are recapitulated by a DnaA_{ts} mutant. CcrZ colocalizes with FtsZ which functions to localize CcrZ and somehow regulate its activity. The data suggest that following initiation the new origins move to the ¼ and ¾ positions and await the arrival of FtsZ/CcrZ. How arrival of

FtsZ/CcrZ licenses replication is still not clear despite efforts to find a direct link between CcrZ and DnaA or YabA. Based on sequence homology CcrA appears to be a phosphotransferase. It is an interesting and novel story and CcrZ appears to be present in all Firmicutes so the findings should be of broad interest even without a mechanism.

One new piece of data is provided that I find confusing. Previously, the authors show strong evidence that FtsZ and CcrZ interact. In this revised version they present data indicating this interaction depends upon the conserved tail of FtsZ (Fig. 3F). I have reservations about this finding although it would be of interest, if confirmed, since it would expand the number of proteins that interact with the tail of FtsZ. I just think it needs confirmation. Various publications have indicated that the interaction between the tail of FtsZ and its partners is extremely weak ($K_d \sim 30\text{-}50 \mu\text{M}$), which is too weak to mediate co-elution (Fig. 3d) (papers from the Lowe lab and Lutkenhaus lab). Also, in 3f (2-hybrid assay), there is presumably no Z ring (since only FtsZ deleted for the tail is expressed). If for some reason the Z ring is necessary for the functional interaction, it would not occur. If the interaction depends upon the tail of FtsZ this could be easily checked *in vitro* (they have the assay for this, just use an FtsZ with the tail deleted).

We thank the referee for this very good point since, as mentioned, the interaction strength between FtsZ-CTD and partners was reported to be very low. We have several experimental data to show that FtsZ full length and CcrZ do interact (Fig. 3). However, our original conclusion, that FtsZ would interact with CcrZ via its C-terminal tail, only depended on an *in vivo* Split Luciferase assay (old Fig. 3f). We have repeated this experiment and indeed confirmed a reduction of signal when using CcrZ-LgBit with FtsZ Δ CTD-SmBit (after depleting FtsZ wild type for as short as 40 min). However, as mentioned by the referee, it would be unlikely if this interaction entirely relies on the C-terminus of FtsZ as we are able to co-purify FtsZ and CcrZ in *E. coli*.

As suggested, we have now performed additional *in vitro* pull-down assays of FtsZ or FtsZ Δ CTD using CcrZ-GFP and indeed, we could still co-purify FtsZ Δ CTD (see figure below, left). In addition, we also have performed a new bacterial 2-hybrid assay using CcrZ and FtsZ full length or the truncated FtsZ Δ CTD and only observed a small decrease in signal when FtsZ-CTD was deleted (see figure below, right). This further supports the pull-down results. These new data suggest that the CcrZ and FtsZ interaction does not depend on the FtsZ-CTD, as originally concluded in our revised manuscript. We have therefore removed this claim from the main text and are very grateful to the referee for pointing this out.

{redacted}

Minor comments:

1. A lot of fluorescent fusions are used that are inserted in the chromosome in place of the WT gene. It seems that these all complement, although it is not specifically stated, *e.g.*, FtsZ-mCherry in Fig. 2. It should be mentioned that cells are normal, and the fusions complement (they look like it).

This is a good point as we use these fusions as markers for our analysis. Throughout our experiments we have verified the growth rates and morphologies of the different -fusion mutants to make sure that they complement fully. In all cases they were like wild type phenotypes. We have therefore added a mention in the text.

2. The suppression of the various phenotypes due to loss of CcrZ by DnaA-S292G and YabA deletion appears to be very good. Are there any noticeable defects in the absence of the CcrZ regulation?

As presented in Fig. 4, the growth rates and morphologies of $\Delta ccrZ$ *dnaA-S292G* or $\Delta ccrZ$ $\Delta yabA$ mutants are like those of a wild type and we could not observe any defect. Indeed, these are interesting observations and are good starting points for follow up research to figure out how initiation of DNA replication is controlled in these suppressors.

3. Line 138. 1:1 ratio. I did not see this data in Fig. 3 or the supplement. I would like to see the sizing column data on the 1:1 ratio of FtsZ to CcrZ; is the size as expected for 1 FtsZ and 1 CcrZ. If the interaction is mediated by the tail of FtsZ this could readily be checked *in vitro*.

There was a miscommunication between authors leading to an incorrect statement, this ratio was in fact not determined by size-exclusion chromatography but by quantifying the bands from SDS-PAGE gel. Since we do not have more accurate information at this point, we have removed the statement of the complex ratio in the revised manuscript.

4. Line 214. Curious if the starting slow growing strain was sequenced? It may have suppressors that aid its growth without producing large colonies.

This is a good point, the strain $\Delta ccrZ$, *hlpA-mKate2* from fig. 4a is a *ccrZ* deletion mutant displaying a slow growth and producing small colonies on agar plates. We have whole genome-sequenced this strain and did not find any mutation, indicating that a $\Delta ccrZ$ can grow without suppressor, but slowly and with DNA defects, as described in Fig. 1. We have uploaded the genome re-sequencing data to SRA (SRX10592634).

5. Line 230. I find it surprising that deletion of CcrZ from *B. subtilis* affected the *ori/ter* ratio but had no effect on morphology, Fig. 1d in extended data.

We observed important growth and morphological defects upon depletion of *ccrZ_{Sa}* in *S. aureus* but could not see any growth defect in *B. subtilis* (Extended Data Fig. 1d). Which was surprising, since the mutant had a clear drop in *oriC/ter* ratios (fig. 4g). As suggested by the referee, we have re-assessed the growth rate of a *B. subtilis* $\Delta ccrZ_{Bs}$ mutant at 37°C, but it was, as mentioned before, like wild type (see figure below top left).

However, we have now carefully examined the cell morphology of hundreds of *B. subtilis* cells (wild type and $\Delta ccrZ_{Bs}$) and noticed a small but significant change in morphology where $\Delta ccrZ_{Bs}$ cells are in fact slightly longer (or less well separated) and thinner (see figure below top right). This observation is consistent with previous work showing that *B. subtilis* cells get longer upon defective DNA replication or DNA damage (Ogura Y, *et al. J Bacteriol.* 2001; Hill NS, *et al. PLoS Genet.* 2012). We have now added this new data to the manuscript (Extended Data Fig. 1d).

To reinforce the hypothesis of defective DNA replication in the $\Delta ccrZ_{Bs}$ mutant, we have compared the growth of *B. subtilis* wild type and $\Delta ccrZ_{Bs}$ in presence of increasing concentrations of ciprofloxacin (fluoroquinolone antibiotics blocking the replication fork). We indeed observed a slight increased susceptibility of the $\Delta ccrZ_{Bs}$ mutant to the antibiotics. Since this result would need to be further confirmed and the assay optimized, we did not add it to the manuscript. Nevertheless, it would be relevant for future studies to characterize in more details this mutant.

6. Line 329. Since you have the purified protein have you checked by mass spec to see if there is a

bound nucleotide?

Identifying a nucleotide bound to CcrZ would be ideal as it could give more information about its functionality. For this purpose, large amount of protein from *S. pneumoniae* would need to be purified and therefore the conditions of production / purification would need to be optimized accordingly. This is an excellent point and future studies will aim at identifying such nucleotide / substrate of CcrZ.

** Reviewer #3 (Remarks to the Author):

The authors have conducted additional experiments in response to my comments and provided additional information regarding the potential mechanism of how CcrZ regulates DnaA protein. Although the mechanism has not been completely clarified, the authors have made effort in their current capacity to gain further insight. I find that the revision is satisfactory, and that the manuscript has been significantly improved that its publication in Nature Microbiology is warranted

Decision Letter, first revision:

Dear Jan-Willem,

Thank you for submitting your revised manuscript "Spatio-temporal control of DNA replication by the pneumococcal cell cycle regulator CcrZ" (NMICROBIOL-21030670A) and for your patience as we waited to hear back from the reviewers. The revised article has now been seen by the original referees and their comments are below. The reviewers find that the paper has improved in revision, and therefore we'll be happy in principle to publish it in Nature Microbiology, pending a few minor revisions to comply with our editorial and formatting guidelines.

We are now performing detailed checks on your paper and all related files, and will send you a checklist detailing our editorial and formatting requirements in about a week. Please do not upload the final materials and make any revisions until you receive this additional information from us.

Thank you again for your interest in Nature Microbiology and in the meantime please do not hesitate to contact me if you have any questions.

With best regards,

{redacted}

Reviewer comments

Reviewer #2 (Remarks to the Author):

The authors have responded to my previous questions, especially clarifying the interaction between FtsZ and CcrZ.

Decision Letter, final checks:

Dear Jan-Willem,

Thank you again for your patience as we've prepared the guidelines for final submission of your Nature Microbiology manuscript, "Spatio-temporal control of DNA replication by the pneumococcal cell cycle regulator CcrZ" (NMICROBIOL-21030670A). Please carefully follow the step-by-step instructions provided in the attached file, and add a response in each row of the table to indicate the changes that you have made. Please also check the attached Word doc with a few marked-up edits we have proposed to the title and abstract. Ensuring that each point is addressed will help to ensure that your revised manuscript can be swiftly handed over to our production team.

In recognition of the time and expertise our reviewers provide to Nature Microbiology's editorial process, we would like to formally acknowledge their contribution to the external peer review of your manuscript entitled "Spatio-temporal control of DNA replication by the pneumococcal cell cycle regulator CcrZ". For those reviewers who give their assent, we will be publishing their names alongside the published article.

Nature Microbiology offers a Transparent Peer Review option for new original research manuscripts submitted after December 1st, 2019. As part of this initiative, we encourage our authors to support increased transparency into the peer review process by agreeing to have the reviewer comments, author rebuttal letters, and editorial decision letters published as a Supplementary item. When you submit your final files please clearly state in your cover letter whether or not you would like to participate in this initiative. Please note that failure to state your preference will result in delays in accepting your manuscript for publication.

Cover suggestions

As you prepare your final files we encourage you to consider whether you have any images or illustrations that may be appropriate for use on the cover of Nature Microbiology.

Nature Microbiology has now transitioned to a unified Rights Collection system which will allow our Author Services team to quickly and easily collect the rights and permissions required to publish your work. Approximately 10 days after your paper is formally accepted, you will receive an email in providing you with a link to complete the grant of rights. If your paper is eligible for Open Access, our Author Services team will also be in touch regarding any additional information that may be required to arrange payment for your article.

Please note that *Nature Microbiology* is a Transformative Journal (TJ). Authors may publish their research with us through the traditional subscription access route or make their paper immediately open access through payment of an article-processing charge (APC). Authors will not be required to make a final decision about access to their article until it has been accepted. [Find out more about Transformative Journals](https://www.springernature.com/gp/open-research/transformative-journals)

Authors may need to take specific actions to achieve compliance with funder and institutional open access mandates. For submissions from January 2021, if your research is supported by a funder that requires immediate open access (e.g. according to [Plan S principles](https://www.springernature.com/gp/open-research/plan-s-compliance)) then you should select the gold OA route, and we will direct you to the compliant route where possible. For authors selecting the subscription publication route our standard licensing terms will need to be accepted, including our [self-archiving policies](https://www.springernature.com/gp/open-research/policies/journal-policies). Those standard licensing terms will supersede any other terms that the author or any third party may assert apply to any version of the manuscript.

When you are ready, please use the following link for uploading all the required documents:

{redacted}

With best regards,

{redacted}

Final Decision Letter:

Dear Jan-Willem,

I am very pleased to accept your Article "CcrZ is a pneumococcal spatio-temporal cell cycle regulator

that interacts with FtsZ and controls DNA replication by modulating the activity of DnaA" for publication in Nature Microbiology. Thank you for having chosen to submit your work to us and many congratulations to you and your co-authors.

Before your manuscript is typeset, we will edit the text to ensure it is intelligible to our wide readership and conforms to house style. We look particularly carefully at the titles of all papers to ensure that they are relatively brief and understandable.

Acceptance of your manuscript is conditional on all authors' agreement with our publication policies (see www.nature.com/nmicrobiolate/authors/gta/content-type/index.html). In particular your manuscript must not be published elsewhere and there must be no announcement of the work to any media outlet until the publication date (the day on which it is uploaded onto our website).

Please note that *Nature Microbiology* is a Transformative Journal (TJ). Authors may publish their research with us through the traditional subscription access route or make their paper immediately open access through payment of an article-processing charge (APC). Authors will not be required to make a final decision about access to their article until it has been accepted. [Find out more about Transformative Journals](https://www.springernature.com/gp/open-research/transformative-journals)

Authors may need to take specific actions to achieve compliance with funder and institutional open access mandates. For submissions from January 2021, if your research is supported by a funder that requires immediate open access (e.g. according to [Plan S principles](https://www.springernature.com/gp/open-research/plan-s-compliance)) then you should select the gold OA route, and we will direct you to the compliant route where possible. For authors selecting the subscription publication route our standard licensing terms will need to be accepted, including our [self-archiving policies](https://www.springernature.com/gp/open-research/policies/journal-policies). Those standard licensing terms will supersede any other terms that the author or any third party may assert apply to any version of the manuscript.

We welcome the submission of potential cover material (including a short caption of around 40 words) related to your manuscript; suggestions should be sent to Nature Microbiology as electronic files (the image should be 300 dpi at 210 x 297 mm in either TIFF or JPEG format). Please note that such pictures should be selected more for their aesthetic appeal than for their scientific content, and that colour

images work better than black and white or grayscale images. Please do not try to design a cover with the Nature Microbiology logo etc., and please do not submit composites of images related to your work. I am sure you will understand that we cannot make any promise as to whether any of your suggestions might be selected for the cover of the journal.

Congratulations once again to you and your co-authors on putting together such a nice story, I look forward to seeing it published.

□